# Identification of Smoke and Sulfuric Acid Aerosol in SAGE III/ISS Extinction Spectra

Travis N. Knepp[1], Larry Thomason[1], Mahesh Kovilakam[2,1], Jason Tackett[1], Jayanta Kar[2,1], Robert Damadeo[1], and David Flittner[1]

[1]NASA Langley Research Center, Hampton, Virginia 23681, USA
[2]Science Systems and Applications, Inc. Hampton, Virginia 23666, USA

**Correspondence:** Travis N. Knepp (travis.n.knepp@nasa.gov)

**Abstract.** We developed a technique to classify the composition of enhanced aerosol layers as either smoke or sulfuric acid aerosol using extinction spectra from the SAGE III/ISS instrument. This method takes advantage of the different spectral properties of smoke and sulfuric acid aerosol, which is manifest in distinctly different spectral slopes in the SAGE III/ISS data. Herein we demonstrate the utility of this method and present an evaluation of its performance using 4 case-study events of 2 moderate volcanic eruptions (2018 Ambae eruption, 2019 Ulawun eruption; both of which released <0.5 Tg of $SO_2$) and 2 large wildfire events (2017 Canadian pyroCb, and 2020 Australian pyroCb). We provide corroborative data from the CALIOP instrument to support these classifications. This method correctly classified smoke and sulfuric acid plumes in the case-study events >81% and >99.5% of the time, respectively. The application of this method to a large volcanic event (i.e., the 2019 Raikoke eruption; $\geq$1.5 Tg $SO_2$) serves as an example of why this method is limited to small/moderate volcanic events as it incorrectly classified Raikoke's larger sulfuric acid particles as smoke. We evaluated the possibility of smoke being present in the stratosphere before and after the Raikoke eruption. While smoke was present during this time period it was insufficient to account for the magnitude of smoke classifications we observed. Therefore, while this method worked well for large-scale wildfire events and eruptions that inject less $SO_2$, the size of the aerosol created by the Raikoke eruption was outside the applicable range of this method.

## 1 Introduction

Stratospheric aerosol consists of submicron particles (Chagnon and Junge, 1961) that are composed primarily of sulfuric acid and water (Murphy et al., 1998) and play a crucial role in atmospheric chemistry and radiation transfer (Pitts and Thomason, 1993; Kremser et al., 2016; Wilka et al., 2018). Background stratospheric sulfuric acid is supplied by chronic, natural, emission of OCS (carbonyl sulfide), $CS_2$ (carbon disulfide), DMS (dimethyl sulfide), and $SO_2$, from both land and ocean sources (Kremser et al., 2016). The total amount of sulfur in the stratosphere is strongly modulated by volcanic activity. In the past few decades, this has most notably been the result of a few events like Pinatubo and El Chichón (McCormick et al., 1995). However, even relatively small events have been shown to impact stratospheric aerosol levels and radiative forcing (Vernier et al., 2011), thus affecting climate and chemistry.

Volcanic eruptions have the potential to significantly change the atmosphere in several ways including changes in chemical composition, atmospheric dynamics, synoptic weather patterns, and radiation transfer. To facilitate comparison of volcanic events from a geological perspective, the Volcanic Explosivity Index (VEI) was developed by Newhall and Self (1982) and was later refined by Pyle (1995) (see Eq. 1 where V is the ejecta volume in cubic meters). While the VEI scale provides meaningful information for geologists (i.e., mass and volume of ejecta as well as rate of ejection), it retains little value for atmospheric scientists who are interested in what was ejected (e.g., $SO_2$, which is converted to sulfuric acid aerosol, as opposed to lava, rocks, and ash) and where the ejected material went (i.e., troposphere vs. stratosphere).

$$VEI = \log_{10}(V) - 4 \tag{1}$$

As an illustration of the ineffectiveness of this scale we consider 2 eruptions: the 1783 Laki eruption and the 1815 Tambora eruption. The Laki eruption (VEI-4) released 120 Tg of $SO_2$ that remained predominantly in the troposphere, but may have been injected as high at 15 km (Thordarson and Self, 2003). This resulted in an exceptionally cold Northern Hemisphere (NH) winter and the Icelandic "haze famine" wherein 20–25% of the Icelandic population and upwards of 60% of the grazing livestock perished (Thordarson and Self, 1993, 2003). The Tambora eruption (VEI-7) released 100 Tg of $SO_2$, much of which went into the stratosphere, which resulted in a global cooling of 2 K and the 1816 "year without a summer" (Stothers, 1984; Schurer et al., 2019). Despite being a significantly smaller eruption (3 orders of magnitude smaller per the VEI scale), Laki released significantly more $SO_2$ into the atmosphere and thereby had a higher potential for impacting atmospheric chemistry and climate. These 2 events serve as an example of the importance of knowing the mass of released $SO_2$ and injection altitude; neither of which are considered in the current VEI scale. The apparent discrepancy between the VEI ratings for Tambora and Laki is attributed to the VEI scale being based primarily on the volume of ejected tephra (solid material) *not* the mass of released gases such as $SO_2$. Therefore, when discussing the size of volcanic eruptions, within the context of this manuscript as well as atmospheric chemistry and climate studies, it is important to bear in mind that the VEI scale fails to be a scale that accurately represents an eruption's potential impact on atmospheric chemistry and radiation transfer. Despite the limitations of the VEI scale, it remains in use by atmospheric and climate scientists until a more meaningful scale is developed, ideally one that takes into consideration the climactic impact of these eruptions.

Probably the best known eruption in recent history is the 1991 eruption of Mount Pinatubo (VEI-6), which ejected ≈20 Tg of $SO_2$, resulted in a global temperature reduction of 0.5–1 K, and changed the stratospheric aerosol levels for 7–9 years (Bluth et al., 1992; Deshler et al., 2003; McCormick et al., 1995; Minnis et al., 1993; Robock, 2002; Santer et al., 2014). Despite being relatively rare, large eruptions have a significant impact on short-term atmospheric chemistry and physics. However, it is not *just* large eruptions that influence atmospheric chemistry and radiative transfer. On the contrary, it has been demonstrated that chronic eruptions of small volcanoes (VEI-3–4) play a measurable role as noted by Vernier et al. (2011), making them relevant to atmospheric chemistry and climate studies. While these eruptions lack the volume of their larger siblings, they make up for the difference in eruption frequency with VEI-3 eruptions taking place 4–5 times per year and VEI-4 eruptions occurring every

1-2 years. This results in a continual injection of $SO_2$, and sometimes ash, into the lower stratosphere and free troposphere, sometimes with devastating consequences.

Another source of stratospheric aerosol that has received increasing attention over the past decade is large-scale, intense-burning, wildfire events that generate pyrocumulonimbus clouds (pyroCbs, also referred to as cumulonimbus flammagenitus), which were originally hypothesized to only exist as a product of nuclear explosions (Turco et al., 1983). These fires burn with sufficient intensity to form a cumulonimbus cloud and inject smoke and volatile organic compounds directly into the stratosphere (Fromm et al., 2006, 2010) on a scale comparable to a VEI-3 or VEI-4 volcanic eruption (Peterson et al., 2018). The largest pyroCbs on record are the 2017 Canadian and 2020 Australian wildfires, both of which injected between 0.1 and 1.0 Tg of aerosol into the lower stratosphere (Peterson et al., 2018; Yu et al., 2019; Kablick III et al., 2020; D'Angelo et al., 2022).

These biomass burning events release black and brown carbon aerosol (BC and BrC, respectively), which is carried into the stratosphere. Unlike sulfuric acid aerosol, BC and BrC absorb solar radiation causing it to heat throughout the day. Due to this diabatic heating, the density of the air mass immediately around the particles decreases and may result in this plume lofting to higher altitudes, well past the initial injection height and independent of the general atmospheric circulation (Yu et al., 2019). This lofting effectively transports the chemical environment present in the lower atmosphere to higher altitudes, which can act as a tracer for pyroCb injections, as was demonstrated for the 2017 Canadian and 2019/2020 Australian wildfires (Boone et al., 2020; Kablick III et al., 2020). Further, displacing the background stratospheric air with air that differs chemically and radiatively can also alter synoptic meteorology (Kablick III et al., 2020). Therefore, while pyroCbs lack the eruptive power of volcanic events, they have the potential to play a substantive role in not only short-term ground-level air quality (Johnston et al., 2020), but also in stratospheric chemistry and dynamics.

Given the broad impact of these two event types, and the fact that they influence the atmosphere in distinctly different ways, it is necessary to be able to distinguish between the two.

A combination of volcanic and pyroCb events occurred in 2019 when Raikoke erupted during the NH's burn season. This coincidence presented both a scientific opportunity as well as measurement challenges. Raikoke injected $SO_2$ and ash to a peak altitude of $\approx$15 km (Thomason et al., 2021) with $SO_2$ detected shortly thereafter at 19 km (Hedelt et al., 2019). This resulted in an $SO_2$ column density in excess of 900 Dobson units (Hedelt et al., 2019) and an overall mass load of $\approx$1.5 Tg (Muser et al., 2020; de Leeuw et al., 2021). This plume was transported to the northeast then down the western seaboard of North America before circling the globe (Hedelt et al., 2019; Chouza et al., 2020; Kloss et al., 2021; Vaughan et al., 2021).

While other instruments such as the Ozone Mapping and Profiler Suite (OMPS), Atmospheric Chemistry Experiment Fourier-Transform Spectrometer (ACE-FTS), the Cloud-Aerosol Lidar with Orthogonal Polarization (CALIOP), and the Tropospheric Monitoring Instrument (TropOMI) are routinely used to observe wildfire and volcanic activity, it is important to understand the applicability of the Stratospheric Aerosol and Gas Experiment III aboard the International Space Station (SAGE III/ISS, hereafter referred to as SAGE) instrument's data record to these identifications. While coincident observations (in both time and space) strengthen the interpretation of each instrument's data, it is not always possible to have multiple instruments observe the same volume of the atmosphere within a reasonable time frame. Therefore, it is necessary to understand the

strengths and limitations of each individual instrument. To this end, we evaluated a method of distinguishing between sulfuric acid aerosol and smoke in the stratosphere that uses the SAGE extinction spectra and discuss limitations of this methodology. The questions addressed in this manuscript are: 1. whether $SO_2$ and smoke can be reliably identified using SAGE's aerosol extinction coefficient data, and 2. if this methodology can be applied to eruptions that are expected to produce large particles because of the large amount of injected $SO_2$ (e.g., the 2019 Raikoke eruption).

## 2 Instrumentation

### 2.1 SAGE III instrument and data preparation

SAGE is a solar and lunar occultation instrument (Cisewski et al., 2014) that is installed on the ISS and has a data record that began in June 2017. The spectrograph sub-system has a spectral range that extends from 280 to 1040 nm and has a resolution of 1-2 nm. In addition to the spectrograph there is an InGaAs photodiode at 1550 nm. The ISS orbit is inclined at $51.6°$, resulting in more observations at midlatitudes than at tropical latitudes as shown by Knepp et al. (2020).

The version 5.2 SAGE data were used in this analysis. The standard products include the number density of gas-phase species for both solar ($O_3$, $NO_2$, and $H_2O$) and lunar ($O_3$, $NO_2$, and $NO_3$) observations, as well as aerosol extinction coefficients (385, 450, 520, 600, 675, 755, 870, 1020, 1550 nm; referenced as $k_\lambda$) for solar occultations. The v5.2 release differed from v5.1 in that vertical smoothing for all products, except $H_2O$, was turned off to provide data at the highest vertical resolution. In this study, the extinction coefficients were filtered to remove data with relative errors in excess of 20% followed by vertical smoothing using a 1-2-1 binomial average to yield a vertical resolution consistent with previous SAGE missions (i.e., 0.75 km, reported every 0.5 km). Finally, the data were limited to altitudes between 2 km above the tropopause (as reported by the Modern-Era Retrospective analysis for Research and Applications, version 2 model) and 30 km.

The 520, 600, and 675 nm extinction coefficients had a low bias as demonstrated by Wang et al. (2020) for the v5.1 product, and this bias remains present in the v5.2 product. The bias is more prominent at mid-latitudes and altitudes between 20 and 25 km, and is likely the product of residual ozone interference in the aerosol retrieval algorithm. Therefore, $k_{520}$ was replaced by applying an Ångström (power law) correction as described by Eq. (2) where $k$ is the extinction coefficient at the subscripted wavelength. The $k_{520}$ channel was part of the original analysis code that was written for this study. However, it was later determined that the analysis could have been carried out without the $k_{520}$ channel with no impact on the interpretation of the results. Therefore, the correction method was included in this manuscript so the reader may understand how we initially intended to mitigate the impact of the low bias and to explain how the extinction coefficient data were prepared for use within the current study.

$$\log k_{520} = \frac{\log\left(\frac{k_{450}}{k_{755}}\right) \cdot \log\left(\frac{520}{755}\right)}{\log\left(\frac{450}{755}\right)} + \log k_{755} \tag{2}$$

## 2.2 CALIOP

The Cloud-Aerosol Lidar with Orthogonal Polarization (CALIOP) instrument is a space-borne elastic backscatter lidar that has been orbiting the earth in the A-train constellation since 2006 (Winker et al., 2010). In September 2018 the orbit was lowered by 16.5 km to correspond to the orbit of CloudSat. The onboard Nd:YAG laser emits polarized radiation at 1064 nm and 532 nm. The total backscatter at 1064 nm and both parallel and perpendicular backscatter at 532 nm provides information on the size and shape of the scattering particles. We used data from the version 4.2 product (L1 for depolarization ratio and L2 for the vertical feature masks), which has improved calibration particularly suitable for stratospheric studies (Kar et al., 2018; Getzewich et al., 2018; Kim et al., 2018). We also used the L3 stratospheric aerosol product, which provides monthly averaged aerosol extinction coefficients and attenuated scattering ratios (i.e., ratio of measured total attenuated backscatter to a modeled molecular backscatter; see Vaughan et al. (2009) for details) in the stratosphere at $5°$(latitude), $20°$(longitude), and 900 m (vertical) resolution (Kar et al., 2019).

## 2.3 ACE-FTS

The Atmospheric Chemistry Experiment (ACE) is a space-borne instrument package that has been orbiting the earth since August, 2003. The primary instrument on ACE is a high-resolution ($\pm25$ cm$^{-1}$) Fourier Transform Spectrometer (ACE-FTS) that collects solar spectra via the occultation technique (vertical resolution is $\approx$4 km Bernath et al. (2005), reported every 0.5 km). For instrument and algorithm details the reader is guided to Bernath et al. (2005) and Boone et al. (2005). While the ACE mission was designed to study, among other things, the impact that biomass burning events have on the troposphere, ACE-FTS data have been used to study the impact of pyroCbs on the lower stratosphere as well (Tereszchuk et al., 2013; Boone et al., 2020). Herein, the v4.2 ACE-FTS products were used to aid in identification of stratospheric smoke after the Raikoke eruption.

## 2.4 TropOMI

The Tropospheric Monitoring Instrument (TropOMI) is part of the European Space Agency's Sentinel-5 Precursor mission that is dedicated to monitoring air pollution from space and has been in orbit since October 2017. The instrument consists of a 2 dimensional spectrometer that yields a swath width of 2600 km, ground pixels sizes as good at 7 km x 3.5 km, and daily global coverage (Veefkind et al., 2012). The retrieval of $SO_2$ (v1.1, L2) is accomplished by using the spectrometer's third spectral band (310–405 nm; algorithm details can be seen in Theys et al. (2017)), which was used herein to identify the location of $SO_2$ plumes after the Raikoke eruption.

## 3 Evaluation of smoke and sulfuric acid extinction spectra from Mie theory

In order to distinguish sulfuric acid aerosol from smoke in a UV/Vis/NIR instrument such as SAGE we must first comprehend the different spectral characteristics of each aerosol type. While the composition and spectral characteristics of smoke are highly variable (Bergstrom et al., 2002; Müller et al., 2005; Park et al., 2018; Kozlov et al., 2014; Liu et al., 2015; Womack

et al., 2021), there is commonality between burning events in that the real component of the refractive index is spectrally flat and the imaginary component is variable (both behaviors being significantly different from sulfuric acid aerosol). Therefore, it is reasonable that the extinction spectra (extinction coefficients or extinction ratios as a function of wavelength) for smoke and sulfuric acid aerosols would differ significantly and that this difference may be useful in distinguishing between the two aerosol types.

As an initial test of this hypothesis, from a theoretical perspective, we used Mie theory to calculate extinction coefficients at SAGE wavelengths for sulfuric acid aerosol and smoke. The primary challenge in carrying out this simulation is the highly-variable nature of smoke's refractive index, which is dependent on fuel source, burn conditions, age, etc. Further, smoke in the stratosphere is aged and its composition has likely changed during its transport by ongoing chemistry (Yu et al., 2019). Therefore, the likelihood of this smoke's refractive index being consistent with the refractive index measured within a labo-
ratory setting or in situ measurements from an aircraft flying in the troposphere is small. To our knowledge there have been no composition or refractive index measurements for stratospheric smoke. An additional complication is the highly variable nature of the BrC refractive index (Alexander et al., 2008; Liu et al., 2015). While the real component of the BrC refractive index has no spectral dependence, previous studies reported refractive indices over a relatively broad range: between 1.3 to 1.9 (Kirchstetter et al., 2004; Müller et al., 2005; Alexander et al., 2008; Chakrabarty et al., 2010; Kozlov et al., 2014; Sumlin
et al., 2018). Further, the imaginary component of the BrC refractive index is wavelength dependent (Kirchstetter et al., 2004; Sumlin et al., 2018) and the range of reported values spans 1-3 orders of magnitude (e.g., see Fig. 4 of Liu et al. (2015) and references therein). Therefore, it is possible that the BrC refractive indices (both the real and imaginary components) may be indistinguishable from BC.

To account for this variability we used two sets of smoke refractive indices to span the range of reasonable refractive index
values. The Bergstrom et al. (2002) refractive indices are representative of BC, and the Sumlin et al. (2018) refractive indices are representative of BrC smoke from biomass burning events (see Table 1 for values). While it is unlikely that stratospheric smoke is composed solely of either BC or BrC (Forrister et al., 2015; Yu et al., 2019) the range of refractive indices used herein effectively covers all potential refractive indices for smoke.

These refractive indices were used to calculate extinction coefficients to model the anticipated extinction spectra. This
simulation consisted of two parts: 1. assume a lognormal distribution with constant mode radius (200 nm; the median radius of a lognormal distribution is commonly referred to as "mode radius" in the aerosol literature and we retain that nomenclature herein) and constant distribution width (1.5) to visualize the expected extinction spectrum (panel (a) of Fig. 1); 2. assume a lognormal distribution with constant distribution width (1.5) and variable mode radius (40–500 nm) to visualize how the spectral slopes (see Sect. 4 for details on slope calculation) changed as a function of particle size and composition (panel (b)
of Fig. 1). This was carried out for particles composed of sulfuric acid as well as varying mixtures of BC and BrC using the refractive indices in Table 1.

At this point we must reiterate that the "real-world" refractive indices for stratospheric BrC may be much closer to the BC values and we present Fig. 1 as representative of a reasonable lower limit for the BrC extinction coefficients and spectral slope. Therefore, given this uncertainty in refractive indices, we explicitly state the caveat that this model is neither intended to be

representative of atmospheric conditions immediately following a pyroCb or volcanic eruption nor is it presented as a predictive model. While the details of the size distribution, refractive indices, and number densities can modulate the differences in the extinction spectra and slope, this has no bearing on the subsequent analysis because the classification methodology was not constrained by the assumptions used in this model. Rather, this model provides a general qualitative understanding of how smoke and sulfuric acid particles influence the extinction spectra at SAGE wavelengths and provide a generalized framework for interpreting the results of this study.

| $\lambda$ (nm) | Sulfuric Acid | BC | BrC |
|---|---|---|---|
| 385 | 1.448 + 0i | 1.75 + 0.50i | 1.55 + 1.0E-2i |
| 450 | 1.434 + 0i | 1.75 + 0.50i | 1.55 + 4.4E-3i |
| 520 | 1.431 + 0i | 1.75 + 0.50i | 1.55 + 2.6E-3i |
| 600 | 1.430 + 6.38E-9i | 1.75 + 0.50i | 1.55 + 2.0E-3i |
| 675 | 1.429 + 1.70E-8i | 1.75 + 0.50i | 1.55 + 2.0E-3i |
| 755 | 1.427 + 7.59E-8i | 1.75 + 0.65i | 1.55 + 2.0E-3i |
| 870 | 1.425 + 1.91E-7i | 1.75 + 0.65i | 1.55 + 2.0E-3i |
| 1020 | 1.421 + 1.51E-6i | 1.75 + 0.75i | 1.55 + 2.0E-3i |
| 1550 | 1.403 + 1.42E-4i | 1.75 + 0.90i | 1.55 + 2.0E-3i |

**Table 1.** Complex refractive indices for smoke and sulfuric acid used in the Mie simulations. The smoke refractive index values were based on data collected by Bergstrom et al. (2002) for BC and Sumlin et al. (2018) for BrC. Sulfuric acid refractive index values are from Palmer and Williams (1975).

The results of this simulation are presented in Fig. 1. The vertical dashed line (panel b) represents the mode radius for background sulfuric acid particles (Deshler et al., 2003), and the shaded region represents the range of expected smoke particle sizes (Moore et al., 2021). Here, it was observed that the extinction spectra (panel a) and spectral slopes (panel b) for BC-containing smoke were consistently different for small particles (i.e., radius <160 nm, panel b). Indeed, within the small particle regime the addition of a small amount of BC significantly changed the slope as compared to BrC or sulfuric acid. However, when sulfuric acid particles are large they are, to a great degree, indistinguishable from smoke unless the smoke is composed of primarily BC (or if the BrC refractive indices are indistinguishable from BC). For example, per Fig. 1 (b), if sulfuric acid particles have a mode radius of 200 nm then they are indistinguishable from smoke particles (90% BrC curve) that have a mode radius of 120 nm. Therefore, while the slopes differ substantially at small particle sizes, this example is a demonstration of how this methodology may be limited: volcanic eruptions that result in large sulfuric acid particles may be indistinguishable from smoke (this is driven by the convergence of extinction coefficients at large particle sizes as demonstrated by Thomason (1992) and Thomason et al. (2008)). This limitation will be explored in the analysis of the Raikoke eruption.

What stood out most in this simulation was the stark contrast between the spectral slopes of the sulfuric acid and smoke aerosol types when the sulfuric acid particles are small. Indeed, the sulfuric acid values changed more rapidly than those for

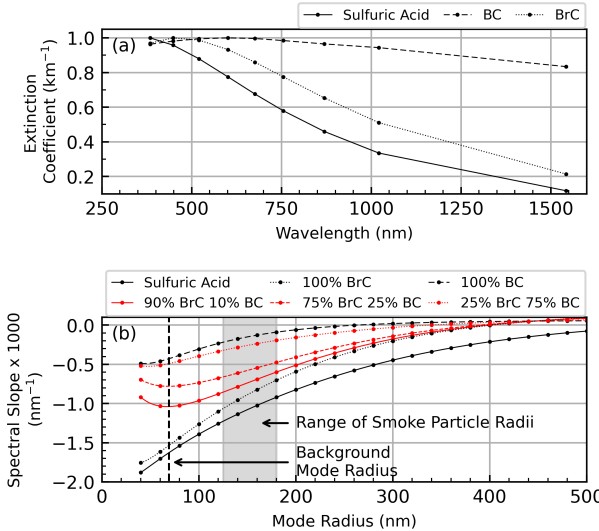

**Figure 1.** Spectra of extinction coefficients (normalized to 1) as a function of wavelength using a fixed mode radius of 200 nm and distribution width of 1.5 (panel a); spectral slope as a function of mode radius using a constant distribution width of 1.5 (panel b; see Sect.4 for details on slope calculation) from the Mie theory simulation. Normalization in panel (a) was carried out by dividing each curve by its maximum value. The vertical dashed line in panel (b) indicates the background mode radius at 20 km, per Deshler et al. (2003). The gray-shaded region in panel (b) indicates a range of smoke particle radii (Moore et al., 2021).

smoke, indicating that, when sulfuric acid aerosol is the predominant aerosol type, the overall slope of the extinction spectrum will be much larger (i.e., more negative) than when the atmosphere is laden with smoke. This distinction provides a testable hypothesis to determine, preliminarily, the viability of separating smoke and sulfuric acid using only extinction coefficients within the UV/Vis/NIR region. To that end, data collected during the four case-study events listed in Table 2 were used to see, broadly speaking, whether the different events showed consistent spectral differences. Data were selected for each event by

truncating the data record to include profiles collected within $\pm 5°$ latitude of the event, and included data collected one month prior to, and three months after, the event (a four-month window) from 14–25 km. The extinction ratios (a proxy for spectral slope) for these four events are presented as a function of $k_{1020}$ in Fig. 2. The two volcanic events (Fig. 2, panels a and b) showed very different behavior from the wildfire events (panels c and d). On one hand, as $k_{1020}$ increased for the volcanic events the extinction ratio increased slightly, though it remained mostly unchanged, suggesting that both the composition and

mean size of the optically important aerosol has remained unchanged from background (following Thomason et al. (2021)). On the other hand, the extinction ratios for the wildfire events had distinctly different behavior, quickly merging to smaller values (<10) as the extinction coefficient increased. This figure demonstrates that the measured extinction ratios behave as expected from the model and that, at least preliminarily, the two event types can be distinguished using SAGE data.

    Up to now we have only considered the raw extinction spectrum (e.g., Fig. 1 (a)) and a simple combination of extinction

coefficients expressed as the extinction ratio (Fig. 2). This was useful for comparing empirical observations to theory and for

| Event | Injected Mass (Tg) | VEI Rating | Date | Latitude |
|-------|--------------------|-----------|------|----------|
| Canadian wildfire | 0.3 Smoke (D'Angelo et al., 2022) | NA | August 2017 | 52°N |
| Ambae eruption | 0.4 $SO_2$ (Malinina et al., 2021) | 3 | April/July 2018 | 15°S |
| Raikoke eruption | 1.5 $SO_2$ (Hedelt et al., 2019) | 4 | June 2019 | 48°N |
| Ulawun eruption | 0.2 $SO_2$ (Kloss et al., 2021) | 4 | June/August 2019 | 5°S |
| Australia wildfire | 1.0 Smoke (D'Angelo et al., 2022) | NA | January 2020 | 35°S |

**Table 2.** Listing of major events used in the current study. All events used data collected between 14 and 25 km. Injected masses refer to the estimated upper limit of total injected mass for each event (not just the mass injected between 14 and 25 km).

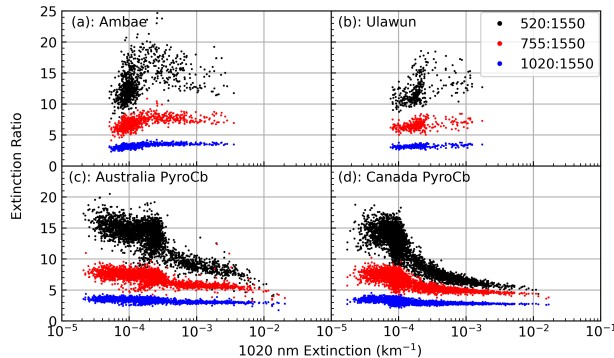

**Figure 2.** Extinction ratio plots, as a function of $k_{1020}$, for 3 ratios in the 4 case study events. All ratios were referenced to the 1550 nm channel and the data were limited to altitudes between 14 and 25 km.

providing rudimentary visualizations, though it requires the analysis to be done on a channel-by-channel or extinction ratio-by-extinction ratio basis (hence the three colors in Fig. 2). Indeed, using a single extinction ratio (e.g., $k_{520} : k_{1020}$) yielded results that were similar to the spectral-slope approach. However, all of the information in these three ratios can be efficiently combined into a single number within the spectral slope, thereby eliminating the channel-by-channel approach, streamlining the analysis, mitigating the potential for noise in a single channel to influence the outcome, as well as mitigating the impact of the low bias in the $k_{520}$ channel. Therefore, given the consistent behavior between the model and the measured extinction ratios we hypothesized that small-to-moderate volcanic events, like the Ambae and Ulawun eruptions, that inject material into the stratosphere can be distinguished from wildfire events in the SAGE record by looking at the extinction coefficients' spectral slope.

## 4  Detection and classification method

To test the aforementioned hypothesis, we evaluated the change in spectral slope as a function of $k_{1020}$ for 4 case-study events (2 pyroCb, 2 volcanic; see Sect. 5 for details). To do this, the spectral slope was calculated via linear regression where channel

wavelength (nm) acted as the independent variable and $\log_{10}(\boldsymbol{k})$ was the dependent variable. The 385 channel was excluded from this analysis because of its rapid attenuation at relatively high altitudes (beginning at $\approx 18$ km). The 600 and 675 nm channels were excluded from the linear regression due to the impact ozone has on these aerosol channels. Further, to reduce the influence of potentially spurious measurements, a conservative cutoff was applied by excluding all extinction coefficients that had relative error >20% and we only used extinction spectra that had valid values in the 6 remaining channels (i.e., 450, 520, 755, 870, 1020, 1550 nm). These slopes were evaluated for each case study event and the results were then applied to data collected after the 2019 Raikoke eruption (see Sect. 6) to evaluate the method's applicability to larger events (compare $SO_2$ injection masses in Table 2).

Not all profiles collected after a volcanic or wildfire event were impacted by that event. Therefore, a method for discriminating between background and perturbed conditions was developed. The use of median ($\tilde{X}$) and median absolute deviation (MAD) has become a popular, statistically robust, alternative to using mean and standard deviation for the elimination of outliers and their impact on an analysis (Leys et al., 2013). If the sample population is normally distributed, then MAD $\cdot$ 1.4826 is equivalent to 1 standard deviation ($1\sigma$). Therefore, we implemented a more rigorous definition of MAD (labeled MAD$^*$, $\cong 2\sigma$) as defined in Eq. (3) where $b = 2 \cdot 1.4826$ and $\boldsymbol{x}$ is an array of the dataset under investigation.

$$\text{MAD}^* = b \cdot \text{median}(|\boldsymbol{x} - \tilde{X}|) \tag{3}$$

Herein, the median and MAD$^*$ of the spectral slopes ($\tilde{X}_m$, MAD$_m^*$) and $k_{1020}$ ($\tilde{X}_k$, MAD$_k^*$) collected during background periods were calculated for each event, as a function of altitude, using data collected within 5°of the each event's latitude. Initially, the background statistics were calculated using only the month prior to each event, but that provided insufficient sampling for the Ambae and Ulawun events due to SAGE's observation schedule (see, for example, Fig. 1 of Knepp et al. (2020)). Therefore, the background time period for the Ambae and Ulawun events was expanded to include 9 months prior to the eruption. The Ulawun eruption required an additional modification. Ambae and Ulawun are geographically close ($\approx 2000$ km) and Ulawun erupted within a year of Ambae's last eruption. Therefore, the stratosphere was still recovering in the months prior to the Ulawun eruption, which biased the background statistics. Background statistics for Ulawun were calculated using profiles collected in Ulawun's latitude band ($\pm 5°$), but using data collected in the 9 months leading up to the Ambae eruption.

Spectra were assigned one of 3 classifications based on the following criteria (here, $\tilde{X}$ and MAD$^*$ refer to background conditions):

1. **Background:** When extinction was not enhanced. i.e.,

$$k_{1020} \leq \tilde{X}_k + \text{MAD}_k^*$$

2. **Sulfuric acid aerosol:** When extinction was enhanced, and the slope was less than (i.e., more negative) or equal to the background slope.

$$(k_{1020} > \tilde{X}_k + \text{MAD}_k^*) \,\&\, (\text{slope} \leq \tilde{X}_m + \text{MAD}_m^*)$$

**3. Smoke:** When extinction was enhanced and the slope was flatter than background conditions.

$$(k_{1020} > \tilde{X}_k + \mathrm{MAD}_k^*) \ \& \ (\mathrm{slope} > \tilde{X}_m + \mathrm{MAD}_m^*)$$

A shortcoming of this classification scheme is that it uses hard cutoff values to separate the aerosol types while, in reality, particles near the smoke/sulfuric acid cutoff would likely be a mixture of the two and not strictly homogeneous. However, the utility of this method, as described, makes the identification of smoke highly conservative.

## 4.1 Layer identification with CALIOP

Ideally, this characterization scheme would be validated with in situ sampling of the various and disparate aerosol layers, which requires expansive sampling on a global (or at least a hemispherical) scale that is not feasible. However, the CALIOP lidar has polarization sensitivity at 532 nm that can be used to make general composition estimates (e.g., sulfuric acid aerosol, smoke, dust, cloud, and volcanic ash). Smoke injected into the stratosphere due to pyroCb events can be discriminated from sulfuric acid aerosol based on the level of depolarization in the return signal (Kim et al., 2018). The ratio of the perpendicular and parallel polarized components of the backscatter (depolarization ratio), provides information about the shape of the scattering particles. In general, the depolarization ratio of tropospheric smoke is quite low (<0.05). However, smoke detected in the stratosphere from pyroCb events has a much higher depolarization ratio (0.1-0.2, per Christian et al. (2020)). This feature can be used to separate stratospheric smoke from the volcanic sulfate particles, which are spherical and have low particulate depolarization ratios (< 0.1). Prata et al. (2017) found mean particulate depolarization ratios of 0.09 and 0.05 for the sulfates from Kasatochi and Sarychev volcanoes. In addition to depolarization ratio, the CALIOP data products contain a vertical feature mask (VFM) product that classifies the different types of detected layers as aerosol (tropospheric and stratospheric) and clouds (Vaughan et al., 2018). Both depolarization ratio and the VFM were used herein to corroborate the identification of sulfuric acid aerosol and smoke within the SAGE data.

## 4.2 Potential for misclassifications in mixed events

In theory, the proposed classification method is straight forward and expected to be reliable for events of a single type (i.e., either volcano or wildfire, but not necessarily mixed events) and when sulfuric acid particles do not become large. However, when both volcanic and wildfire events occur within the same time frame and latitude band the chance for mixing is high, leading to ambiguous determinations of composition. For example, the method presented in this manuscript only considers the integrated properties of the identified aerosol layers and cannot partition out the relative contribution of each aerosol type. Therefore, if smoke mixes with a sulfuric acid plume then we observe the total extinction of that layer and, based on the corresponding slope of the extinction coefficient spectrum, that layer is classified as either smoke or sulfuric acid; this is a false dichotomy.

Figure 3 demonstrates how the spectral slope can change as a function of smoke fraction, relative to pure sulfuric acid aerosol. This simulation was carried out for 2 sulfuric acid distributions to represent 2 scenarios: 1. smoke was injected into the background stratosphere, in the absence of recent volcanic activity (mode radius of sulfuric acid aerosol was set to 70 nm); 2.

smoke was injected into the stratosphere when sulfuric acid particle sizes were larger because of recent volcanic activity (mode radius of sulfuric acid aerosol was set to 150 nm). In both of these scenarios the smoke particle mode radius was fixed at 200 nm, and all distributions had widths of 1.5. This was carried out for 5 different smoke compositions that had varying degrees of BC content, similar to Fig. 1, thereby showing how the smoke composition may influence the magnitude of the relative change

in slope with BrC having the smallest impact and BC having the largest, in agreement with Fig. 1. For example, according to Fig. 3, when 1% of the particles are smoke (99% are sulfuric acid particles) that is injected into a previously-perturbed stratosphere (i.e., when the mode radius of sulfuric acid particles is 150 nm) then the overall spectral slope will be changed by ≈1.8–7% as compared to pure sulfuric acid. If, on the other hand, the same smoke were injected into the stratosphere under background conditions (i.e., the mode radius of sulfuric acid particles is 70 nm) then the overall spectral slope may be changed

by 35–60%.

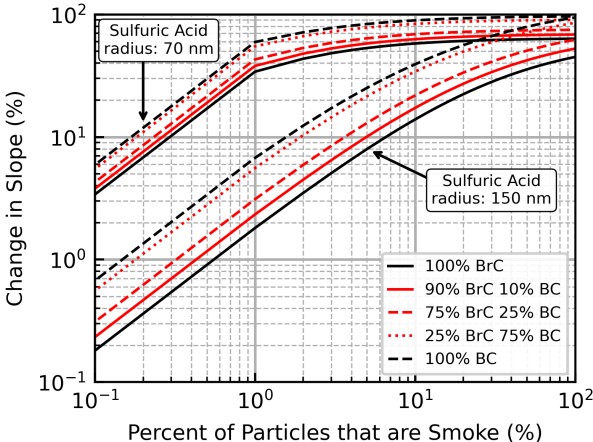

**Figure 3.** Change in spectral slope, relative to pure sulfuric acid aerosol, as a function of smoke fraction for 5 smoke compositions. Here, when 10% of the particles are smoke, the remaining 90% are sulfuric acid. The smoke particles had mode radius of 200 nm and distribution width of 1.5. The sulfuric acid distributions had a width of 1.5 and mode radii of 70 nm and 150 nm.

Again we recognize the multiple assumptions built into this model, much like Fig. 1, and we consider this to be a general guide for interpreting the analysis of mixed events. While the events considered within this manuscript were expected to be single-sourced, the information from Fig. 3 may be useful for interpreting potential misclassifications of smoke. Therefore, Figs. 1 & 3 demonstrate that a relatively small fraction of BC-containing smoke can significantly flatten the spectral slope.

This change in slope is larger when the sulfuric acid particle size is at background levels (i.e., ≈70 nm per Deshler et al. (2003)), which may be sufficient to push these values into the smoke classification. This distinction is important because the method we present considers the change in slope, relative to background conditions, and does not rely solely on post-eruption/pyroCb extinction ratios (per previous studies) or slopes. Therefore, this figure will be used in the upcoming analysis to better understand potential misclassifications within the case-study and Raikoke events.

## 5 Application to case studies events

We considered five events that had significant impact on the stratosphere as detailed in Table 2. All events were classified as either primarily volcanic or pyroCb related. The Ambae and Ulawun eruptions were similar in that they released relatively small amounts of $SO_2$ (0.2–0.4 Tg) as compared to Raikoke (1.5 Tg) and were expected to form smaller particles than Raikoke. Since Raikoke was distinctly different it was sequestered from the list of case-study events and was evaluated separately (Sect. 6). This left four case-study events (2 volcanic, 2 pyroCb) for evaluating distinct behaviors for each event class. While the majority of the data collected for these events appears to come from a single source, we add the caveat that some events were close enough in time and geography to experience some carryover (e.g., the two Ulawun eruptions and the Australian pyroCb), which will be briefly discussed below.

To better appreciate the finer details of the profile data, and to demonstrate which parts of the atmosphere were most impacted by each event, the data were broken into 1 km bins. Statistics for labeling the different layer types in the 4 case-study events are presented in Table 3, which contains the total number of valid spectra collected at each altitude, the number of non-background spectra identified using the above cutoff criteria (Sect. 4), and the fraction of enhanced spectra identified as either smoke or sulfuric acid aerosol. Because the case-study events were predominantly single-sourced, asterisks were inserted in Table 3 to identify probable misclassifications.

### 5.1 Volcanic events

The classification scheme worked well for both volcanic case-study events as there were only 34 spectra mis-classified as smoke (out of >6400, ≈0.5%). It was observed that as extinction increased, the slopes tended to remain approximately consistent with background slopes, or became slightly more negative as seen in Figs. 4 and 5. This moderate decrease in slope (i.e., shifting to more negative values) may be due to the formation of smaller particles immediately after the eruption. Indeed, Fig. 1 shows that smaller sulfuric acid particles should lead to more negative slopes. Overall, there was little deviation from background conditions other than the enhanced extinction coefficient. This limited change in slope is reasonable because background stratospheric aerosol is composed of primarily sulfuric acid and the injection of $SO_2$ from moderately-sized volcanic events led to further formation of sulfuric acid aerosol. While this increased the overall extinction coefficient, its impact on the spectral slope was minimal due to the consistent composition, lack of significant change in the particle size distribution and hence spectral properties under background and elevated loads.

We note that the Ulawun event had a small number of points that were classified as smoke with elevated $k_{1020}$ between 17 and 18 km. While it is not unreasonable to have statistical anomalies, we note that these spectra were collected in January and February 2020, at the peak of the Australian wildfire, when smoke had transported over the Ulawun latitude range (Kloss et al., 2021). While we cannot definitively attribute these data points to smoke from the Australia pyroCb, the general pattern observed here (slope rapidly approached 0 with increasing extinction) is consistent with what was observed for the pyroCb events (*vide infra*) and we note this as interesting.

| | | 14 | 15 | 16 | 17 | 18 | 19 | 20 | 21 | 22 | 23 | 24 | 25 |
|---|---|---|---|---|---|---|---|---|---|---|---|---|---|
| | | | | | | | Altitude (km) | | | | | | |
| Ambae Eruption | Total Spectra | 4 | 15 | 41 | 148 | 628 | 1391 | 1583 | 1586 | 1586 | 1586 | 1586 | 1586 |
| | Enhanced Layers | 0 | 3 | 17 | 51 | 249 | 479 | 479 | 339 | 189 | 48 | 8 | 0 |
| | Sulfuric Acid | — | 1.0 | 1.0 | 1.0 | 1.0 | 1.0 | 1.0 | 1.0 | 0.97 | 0.69 | 0.38 | — |
| | Smoke | — | — | 0 | 0 | 0 | 0 | 0 | 0 | *0.03 | *0.31 | *0.62 | — |
| Ulawun Eruption | Total Spectra | 0 | 1 | 2 | 29 | 205 | 755 | 889 | 892 | 892 | 892 | 892 | 892 |
| | Enhanced Layers | 0 | 1 | 2 | 18 | 154 | 627 | 867 | 883 | 855 | 704 | 395 | 34 |
| | Sulfuric Acid | — | 0 | 1.0 | 0.89 | 0.97 | 1.0 | 1.0 | 1.0 | 1.0 | 1.0 | 1.0 | 1.0 |
| | Smoke | — | *1.0 | 0 | *0.11 | *0.03 | 0 | 0 | 0 | 0 | 0 | 0 | 0 |
| Canadian Fire | Total Spectra | 2690 | 3194 | 3386 | 3546 | 3706 | 3794 | 3800 | 3800 | 3800 | 3800 | 3800 | 3800 |
| | Enhanced Layers | 2204 | 2669 | 2868 | 2957 | 2916 | 2808 | 2298 | 1689 | 1254 | 1267 | 1327 | 1217 |
| | Sulfuric Acid | *0.62 | *0.27 | *0.20 | *0.19 | *0.16 | *0.20 | *0.12 | *0.13 | *0.26 | *0.57 | *0.90 | *0.99 |
| | Smoke | 0.38 | 0.73 | 0.80 | 0.81 | 0.84 | 0.80 | 0.88 | 0.87 | 0.74 | 0.43 | 0.10 | 0.01 |
| Australian Fire | Total Spectra | 3402 | 3789 | 3935 | 4029 | 4113 | 4135 | 4138 | 4138 | 4138 | 4138 | 4138 | 4138 |
| | Enhanced Layers | 2472 | 2885 | 3080 | 3204 | 3218 | 3142 | 2885 | 2710 | 2620 | 2441 | 2172 | 1865 |
| | Sulfuric Acid | *0.01 | *0.01 | *0 | *0.01 | *0.04 | *0.06 | *0.06 | *0.06 | *0.09 | *0.14 | *0.14 | *0.40 |
| | Smoke | 0.99 | 1.0 | 1.0 | 0.99 | 0.96 | 0.94 | 0.94 | 0.94 | 0.91 | 0.86 | 0.86 | 0.60 |

**Table 3.** Layer classification statistics from the SAGE data for the 4 case-study events. Total number of valid spectra, total number of identified layers as well as the fraction of spectra identified as smoke or sulfuric acid aerosol for each case-study event. These case-study events were single sourced, so the presence of smoke in volcanic events or enhanced sulfuric acid in wildfire events is expected to be negligible and would be considered misclassification. These potential misclassifications were labeled with an asterisk (*) for clarity.

Though evidence for ash in the stratosphere, for these events, is tenuous, we cannot categorically exclude the possibility that stratospheric ash was present for at least part of each event's time period. However, if ash were present, it would result in more spectra being classified as smoke because, as discussed above, large particles tend to flatten the extinction spectra while enhancing $k$. Indeed, this may be the cause of some of the smoke classifications in the Ulawun event mentioned above. However, particles of this size are not expected from these moderately-sized eruptions and ash was not identified in the CALIOP VFM. Therefore, we conclude that ash and any other potentially large aerosol (sulfuric acid) did not appreciably impact the optical measurements and that the majority of slopes presented in Figs. 4, 5 are reflective of small sulfuric acid aerosol only.

Figures 6 and 7 show examples of the CALIOP total attenuated backscatter at 532 nm ($\beta_{532}$; panel a), depolarization ratio (panel b), the CALIOP VFM (panel c), SAGE extinction profiles (panel d) as well as the SAGE spectral slope profile (panel e)

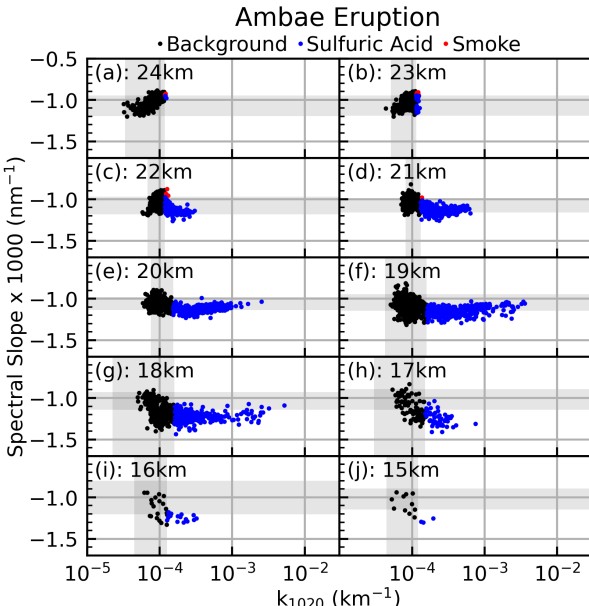

**Figure 4.** Spectral slope (x1000) as a function of $k_{1020}$ and altitude for data collected after the Ambae eruption and within the Ambae latitude band. The gray shaded regions indicate the width of MAD$^*$ as described in Eq. (3).

for the Ambae and Ulawun eruptions, respectively. The figure title provides the SAGE overpass date, latitude, longitude, and distance to the nearest CALIOP profile. The vertical arrow above panel (a) indicates the location of the SAGE overpass relative to the CALIOP curtain plot. For both events, the peak in extinction corresponded well with a rapid decrease in spectral slope (i.e., became more negative) and a stratospheric aerosol layer identified in the CALIOP VFM between 17 and 19 km, and no significant depolarization, giving credence to the SAGE-based identification scheme for sulfuric acid aerosol.

## 5.2 Wildfire events

The wildfire events showed a mix of classifications, though the classification became uniformly smoke with increasing $k_{1020}$, as shown in Figs. 8 & 9. The lowermost altitudes for the Australia events showed a nearly monolithic identification of smoke, as well as a distinct separation from background conditions, which makes sense since the lowermost altitudes are the most impacted by these events. The Canadian wildfire likewise showed a nearly uniform identification of smoke at elevated $k_{1020}$. One thing that stands out in these figures is the overall change in slope as compared to background conditions. These slopes changed by up to 80%, which points toward 2 interpretations: 1. the smoke particles were composed of primarily BC (Fig. 3), contrary to Yu et al. (2019); 2. the smoke particles were composed of BrC but the BrC refractive indices (both the real and imaginary components) were higher than indicated by Sumlin et al. (2018) (i.e., they were similar to BC).

While the volcanic events showed nearly uniform identification of sulfuric acid aerosol under elevated conditions, the wildfire events showed a significant portion of the spectra identified as sulfuric acid aerosol ($\approx$10,000 out of >58,000; $\approx$19%). Of

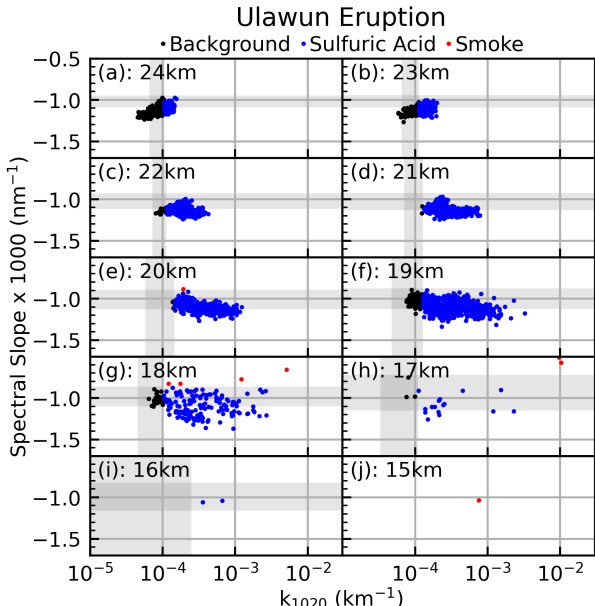

**Figure 5.** Same as Fig. 4, but for the Ulawun eruption.

these ≈10,000 sulfuric acid classifications, ≈48% of them were in the Canadian wildfire event within a narrow altitude range (23–25 km). The reason for the potential presence of elevated sulfuric acid aerosol within the wildfire events could be for two reasons. First, within the current identification scheme, there is the possibility that smoke may be misclassified as sulfuric acid because of the combination of the SAGE viewing geometry and the optical thinness of parts of the smoke plume (i.e., depending on whether SAGE is sampling through the centroid of the plume or only the outer edge where the smoke fraction is low). It is reasonable that, when sampling optically thin smoke layers, the extinction will be elevated above background levels, but the slope may not deviate significantly. This may be achieved when the viewing geometry is such that the path length through an optically thin portion of the smoke plume is sampled, thereby raising the extinction coefficients, but the spectral slope remains effectively unchanged because of the paucity of smoke particles within this layer. This may lead to an ambiguous characterization of aerosol composition at extinctions that are outside background values, but still at the lower end of extinction values for that particular event as seen in Figs. 8 & 9. This scenario is the most likely and is not unexpected from a statistical viewpoint. Secondly, a less likely scenario is there may be elevated levels of sulfuric acid aerosol within the sampling volume due to transport from a nearby and recent volcanic event. Indeed, this may be the case for some of the spectra classified as sulfuric acid in the Australian wildfire case study. In contrast to the Canadian wildfire, the range of background extinction coefficients for the Australian fire spanned a wider range and extended into higher extinction coefficients (e.g., >5E-4 at 16 km), indicating the apparent background conditions were perturbed, potentially from the 2019 Ulawun eruptions (Kloss et al., 2021). If this is the case, then it is reasonable for the sulfuric acid particles to be larger than they would be under background conditions, and the potential impact this has on the change in spectral slope can be seen in Fig. 3. Regardless of why spectra

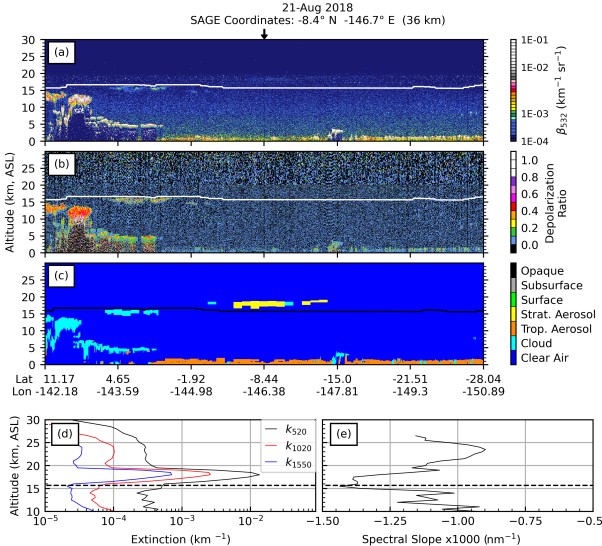

**Figure 6.** CALIOP and SAGE data collected during the Ambae eruption. The title indicates the SAGE profile date, SAGE overpass coordinates, and the distance between the SAGE profile and the nearest CALIOP profile. Panel (a) presents the CALIOP L1 total attenuated backscatter at 532 nm; Panel (b) presents the CALIOP L1 depolarization ratio; Panel (c) presents the CALIOP L2 VFM; Panel (d) presents the SAGE extinction coefficient profiles at 3 wavelengths; and Panel (e) presents the spectral slope profile of the SAGE extinction spectrum. The vertical arrow above panel (a) indicates the location of the SAGE profile along the CALIOP flight path. Solid horizontal lines (panels a, b, and c) and dashed horizontal lines (panels d and e) indicate tropopause altitude. The CALIOP granule used to generate this image is CAL_LID_L1-Standard-V4-10.2018-08-21T11-04-30ZN.

were classified as sulfuric acid, the performance of this identification scheme remains encouraging as the majority (>81%) of non-background values were identified as smoke within the enhanced layers as shown in Table 3. Further, the distribution of data for the wildfire events is markedly different from the volcanic events, indicating that we observed two distinctly different aerosol types.

Figures 10 and 11 show examples of the CALIOP and SAGE profile data collected over the two wildfire case-study events. Here, CALIOP showed significant depolarization near 19 km (Canadian pyroCb) and 14 km (Australian pyroCb), which corresponded well with a rapid increase in both aerosol extinction and spectral slope in the SAGE profile data. We note that in Fig. 11 SAGE saw another layer at 19 km that was not manifest within the CALIOP data. This altitude is well within the SAGE instrument's operational altitude range and may be reflective of the relatively poor return signal at this altitude for CALIOP and its narrow swath width. Alternatively, SAGE may have sampled a narrow smoke filament that was not within the CALIOP sample volume.

Overall, the CALIOP data products provided good support for the SAGE-based classification of stratospheric aerosol composition for case studies that were unambiguously single-sourced and when the volcanic events were moderate in size. Therefore, this identification scheme will now be applied to a more complicated event: the 2019 Raikoke eruption.

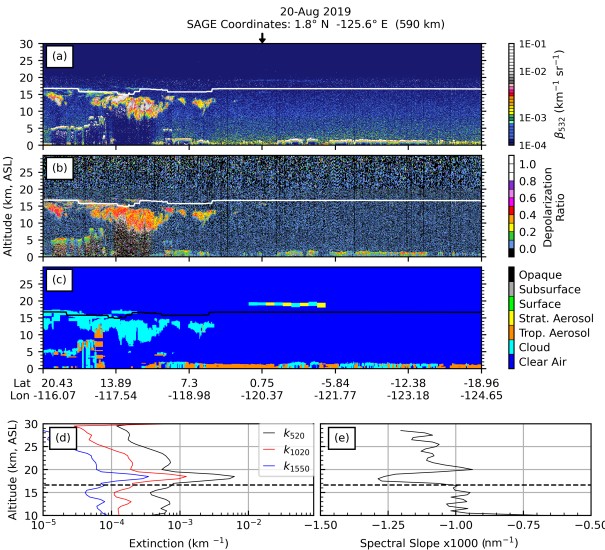

**Figure 7.** Same as Fig. 6, but for the Ulawun eruption. The CALIOP granule used to generate this image is CAL_LID_L1-Standard-V4-10.2019-08-21T09-34-46ZN.

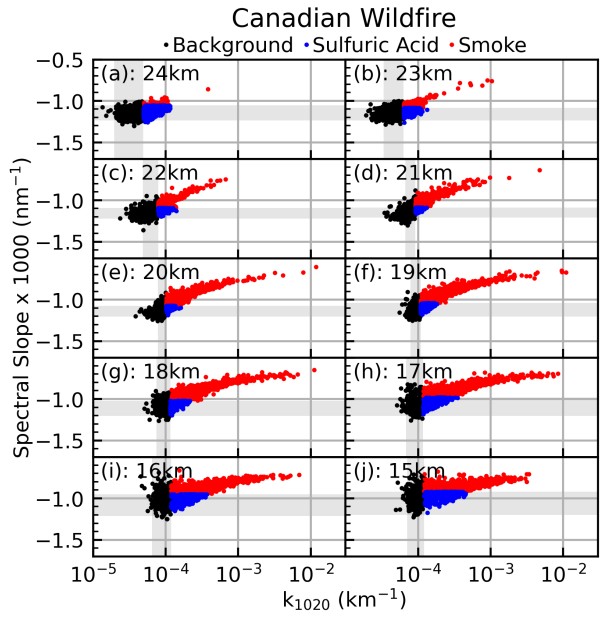

**Figure 8.** Same as Fig. 4, but for the Canadian wildfire.

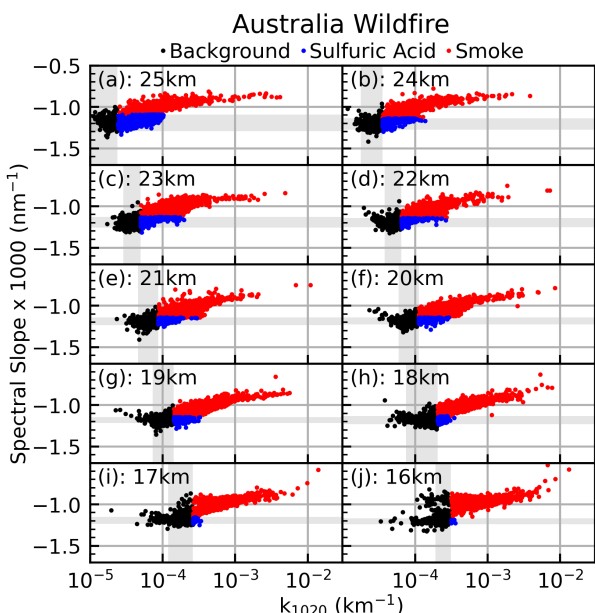

**Figure 9.** Same as Fig. 4, but for the Australian wildfire.

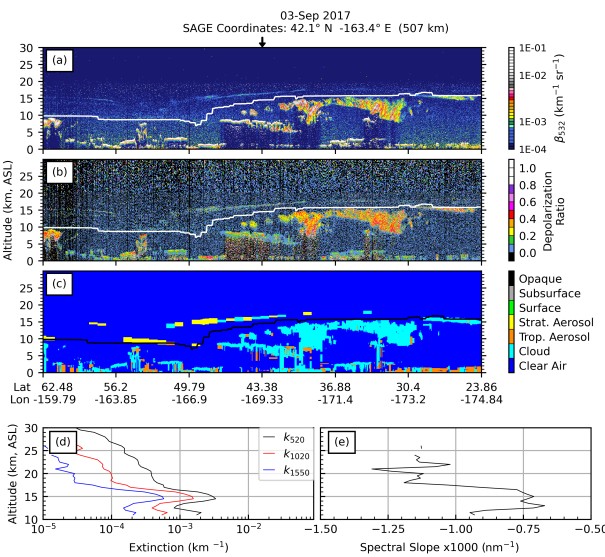

**Figure 10.** Same as Fig. 6, but for the Canadian pyroCb. The CALIOP granule used to generate this image is CAL_LID_L1-Standard-V4-10.2017-09-04T13-24-46ZN.

## 6 Application to the Raikoke event

Located on the Kuril archipelago, Raikoke (48.3°N, 153.3°E) is a volcanic island that has a history of moderately-sized eruptions that have been recorded over the last 200 years (Tanakadate, 1925; Newhall and Self, 1982; Rashidov et al., 2019).

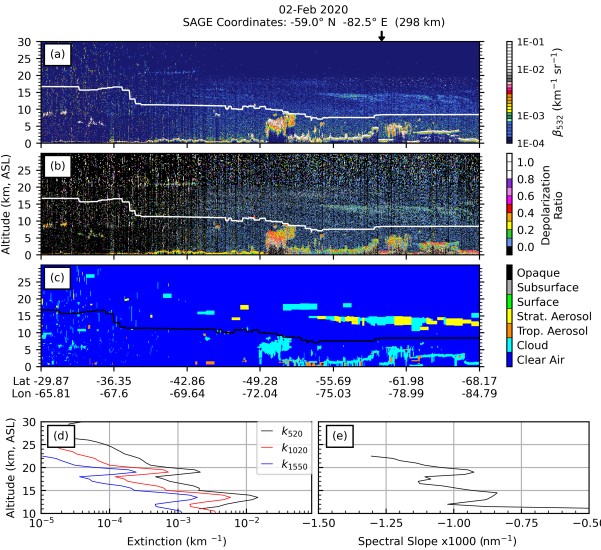

**Figure 11.** Same as Fig. 6, but for the Australia pyroCb. The CALIOP granule used to generate this image is CAL_LID_L1-Standard-V4-10.2020-02-02T05-24-58ZN.

The 22-June, 2019 eruption was the largest volcanic eruption since 2011 (Puyehue-Cordón Caulle) and injected ≈1.5 Tg of sulfur dioxide ($SO_2$) into the lower stratosphere (Muser et al., 2020; de Leeuw et al., 2021); substantially more than the Ambae and Ulawun eruptions combined (Table 2). Therefore, despite having a VEI-4 rating (the same as Ulawun) it was expected to have a more significant impact on atmospheric chemistry and radiation budget. While the injected $SO_2$ mass, by itself, is insufficient to predict particle size and impact on chemistry and climate, we use this parameter as a general guide to distinguishing between "smaller" and "larger" eruptions. Further, while the Raikoke eruption is interesting by itself, the time period surrounding the eruption is of particular interest because of coincident wildfires in the northern hemisphere (Kloss et al., 2021; Vaughan et al., 2021), as well as the increased frequency of volcanic and large-scale wildfire events over the previous decade (e.g., Fromm et al., 2010; Kablick III et al., 2020; Kloss et al., 2020, 2021).

In addition to injecting more $SO_2$, Raikoke also injected a substantial amount of ash directly into the stratosphere, at around 15 km altitude, with $SO_2$ observed at 19 km shortly thereafter (Hedelt et al., 2019). Immediately after the eruption, the primary plume broke into two distinct plumes. One plume moved southward and appeared to be primarily ash (Kloss et al., 2021; Vaughan et al., 2021) that settled out within a week of the eruption (Kloss et al., 2021). The second plume, however, moved to the north and east and was composed primarily of $SO_2$ (Kloss et al., 2021), which was in the process of being converted into sulfuric acid. While the bulk of this $SO_2$ plume remained between Kamchatka and Alaska for 2-3 weeks (de Leeuw et al., 2020; Vaughan et al., 2021) a narrow $SO_2$ filament was observed over North America by the SAGE, CALIOP, and TropOMI instruments within 10 days of the eruption as shown in Fig. 12. The enhanced extinction coefficient in the SAGE profiles

corresponded to the increased backscatter coefficient in CALIOP (12–13 km), both of which coincided with the $SO_2$ filament
observed by TropOMI (Fig. 12, panel b).

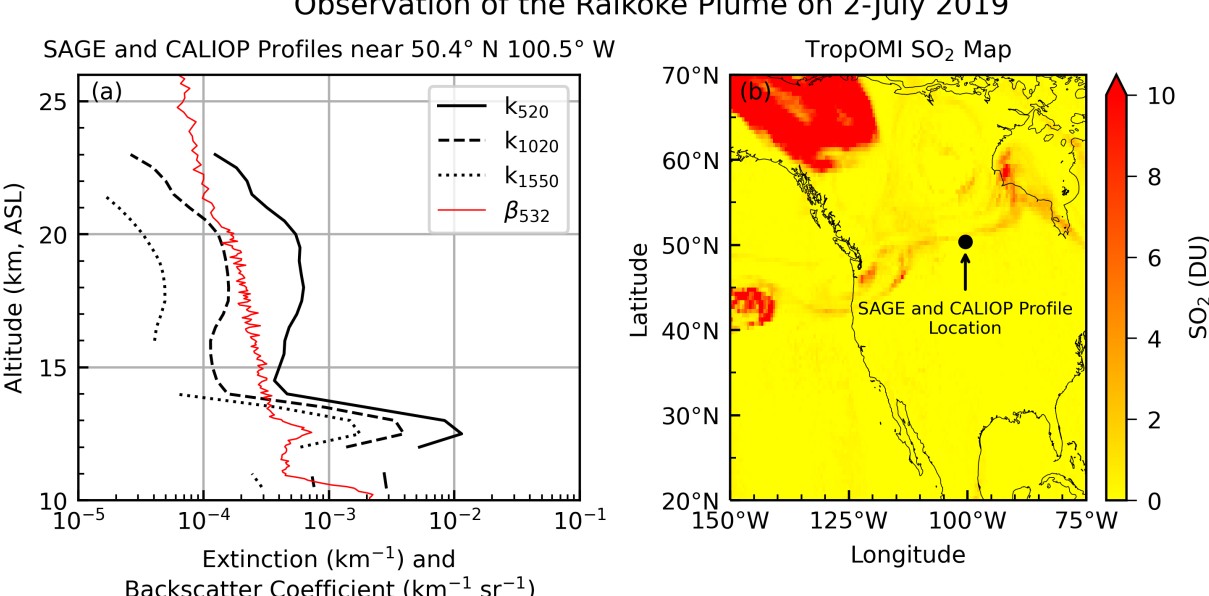

**Figure 12.** Observation of the Raikoke plume on 2-July 2019. SAGE extinction coefficient profiles at 3 wavelengths and CALIOP L1 backscatter profile showing the Raikoke plume between 12 and 14 km near $50.4°$ N and $100.5°$ W (panel a). Breaks in the SAGE profiles is where the algorithm failed to produce valid data. The TropOMI $SO_2$ product shows a wisp of the $SO_2$ plume that intersected the SAGE and CALIOP fields of regard (panel b). The CALIOP granule used in this image was CAL_LID_L1-Standard-V4-10.2019-07-02T09-10-53ZN. The tropopause at this location was below 10 km.

Figure 13 shows the monthly zonal mean extinction coefficient at 1550 nm ($k_{1550}$) and extinction ratio between the 520 nm and 1550 nm channels ($k_{520} : k_{1550}$) from SAGE as well as the attenuated scattering ratio from CALIOP. The progression of enhanced extinction is seen in panels a-f of Fig. 13. Beginning in July, the extinction coefficient increased between 11 and 13 km and this increase was attributed to Raikoke. No significant enhancement was observed in June because these figures present
monthly zonal means and the eruption occurred late in the month, effectively averaging out any enhancement that was detected in the SAGE data. Subsequent months showed significant enhancement as well as how this enhanced layer was transported southward, which is better seen in the extinction ratio plots (panels g-l) and attenuated scattering ratio from CALIOP (panels m-r).

What stood out in Fig. 13, panel (d), was the presence of an enhanced layer at $\approx 23$ km, near $25°N$. The initial ascent of this
"secondary plume" might be seen as early as August (panel c) and remained visible in the extinction coefficient plots for the remainder of the year. The extinction ratio plots (panels g-l) as well as the attenuated scattering ratio plots (panels m-r) more readily show the persistence of this layer through November.

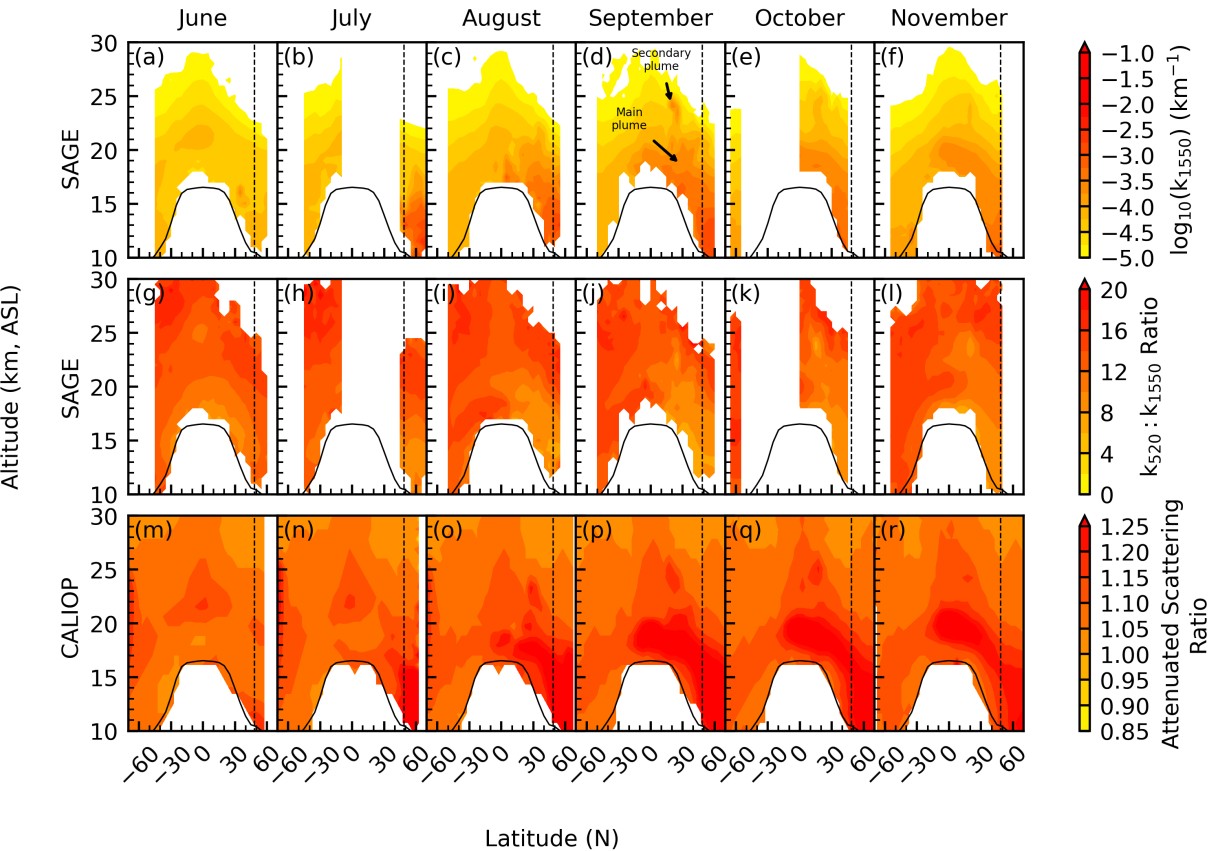

**Figure 13.** Zonal monthly mean of 1550 nm extinction coefficients (panels a-f), 520:1550 extinction ratio (panels g-l), and attenuated scattering ratio from CALIOP (panels m-r). The solid black line indicates the average tropopause altitude and the dashed black line indicates Raikoke's latitude.

Similar to the case-study events, the classification method was applied to data collected near Raikoke's latitude (48°N, ±5°) within 7 months after the eruption. Despite limited NH pyroCb activity, Raikoke was expected to behave as a single-source event as the 2019 pyroCbs were significantly smaller than the Canadian and Australian fires. Because of the broad time range and the lack of longitudinal limitations there were a non-negligible number of spectra that fell into the background classification, similar to the case-study events.

## 6.1 Interpretation of the primary Raikoke plume

Based on the results presented in the previous section, we anticipated that the spectral slopes of the SAGE data collected over the Raikoke event would behave similar to those for Ambae and Ulawun. However, as shown in Fig. 14 and Table 4 the Raikoke data presented what appears to be a mixture of sulfuric acid aerosol and smoke, with the predominant classification being smoke

(only 10-30% of spectra were identified as sulfuric acid aerosol). Indeed, the majority of lower-altitude spectra were identified as smoke, while the balance shifted to sulfuric acid at the highest altitude (23 km). While we anticipated observing some smoke within the profiles, because of the limited NH pyroCb, we did not expect the majority of the spectra to be identified as such. Based on this general classification scheme the Raikoke data looked more like a wildfire event than a volcanic event. Given the magnitude of this eruption, the spectra identified as smoke here may be the product of ash and/or large particle formation. A discussion on reasons for misclassifications is presented in Sect.6.4.

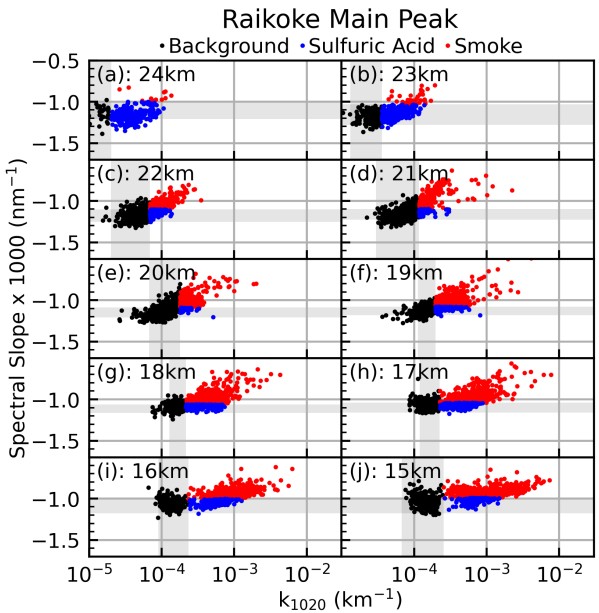

**Figure 14.** Same as Fig. 6, but for the Raikoke eruption.

## 6.2 Interpretation of the secondary Raikoke plume

As shown in Fig. 13, a secondary layer of elevated aerosol broke off from the primary Raikoke plume as it moved southward and continued to loft to higher altitudes. This layer continued to circle the globe before it reached a maximum altitude between 23 and 25 km between $10°$ and $25°$N (Chouza et al., 2020). The classification method was applied to the SAGE extinction spectra that were collected between $10°$N and $30°$N (within 7 months of the eruption). The results of the altitude-based classification are presented in Fig. 15 as well as the statistics in Table 4.

We observed distinctly different patterns between the case-study event types (i.e., slightly decreasing or near-constant slope with increasing extinction for volcanic events (Figs. 4, 5) and flattening of slope with increasing extinction for wildfire events (Figs. 8, 9)). Both of these general patterns were observed in profiles used in creating Fig. 15, sometimes at the same altitude. For example, the data at 19 and 20 km showed both patterns, which led to a bifurcation in slope at higher extinction coefficients.

| | | | | | | | Altitude (km) | | | | | | |
|---|---|---|---|---|---|---|---|---|---|---|---|---|---|
| | | 14 | 15 | 16 | 17 | 18 | 19 | 20 | 21 | 22 | 23 | 24 | 25 |
| **Raikoke Primary** | Total Spectra | 1231 | 1483 | 1590 | 1631 | 1646 | 1652 | 1652 | 1652 | 1652 | 1652 | 1652 | 1652 |
| | Enhanced Layers | 969 | 1166 | 1236 | 1211 | 1078 | 753 | 329 | 344 | 389 | 387 | 261 | 123 |
| | Sulfuric Acid | 0.11 | 0.26 | 0.34 | 0.26 | 0.25 | 0.16 | 0.11 | 0.20 | 0.35 | 0.91 | 0.95 | 0.96 |
| | Smoke | *0.89 | *0.74 | *0.66 | *0.74 | *0.75 | *0.84 | *0.89 | *0.80 | *0.65 | *0.09 | *0.05 | *0.04 |
| **Raikoke Secondary** | Total Spectra | 10 | 32 | 69 | 185 | 458 | 812 | 896 | 896 | 896 | 896 | 896 | 896 |
| | Enhanced Layers | 10 | 28 | 54 | 127 | 319 | 484 | 471 | 380 | 60 | 108 | 218 | 125 |
| | Sulfuric Acid | 0 | 0.5 | 0.56 | 0.65 | 0.70 | 0.71 | 0.69 | 0.72 | 0.20 | 0.02 | 0.33 | 0.70 |
| | Smoke | *1.0 | *0.5 | *0.44 | *0.35 | *0.30 | *0.29 | *0.31 | *0.28 | *0.80 | *0.98 | *0.67 | *0.30 |

**Table 4.** Same as Table 3 but for the Raikoke event. As discussed in the main text, the smoke classifications within this table are most likely misclassifications of large sulfuric acid particles and/or ash. For distinction between Raikoke Primary and Raikoke Secondary refer to Sect.6.1. Because of the lack of support for extensive stratospheric smoke during the Raikoke time period the smoke classifications were flagged as being highly probable misclassifications as indicated by the asterisk (*).

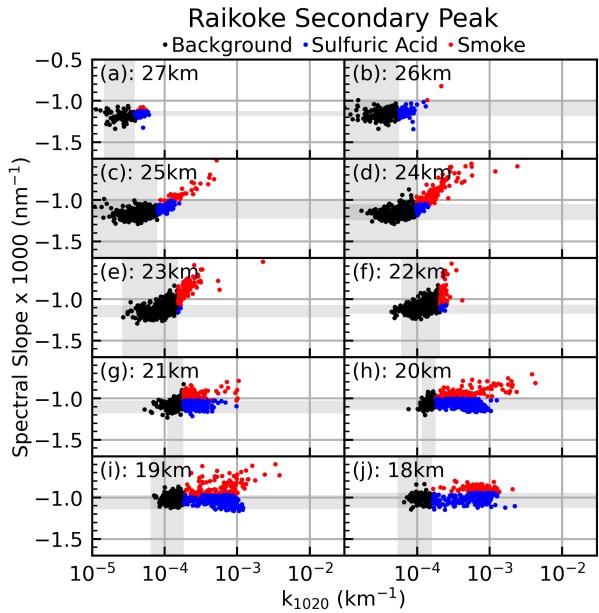

**Figure 15.** Same as Fig. 4, but for the elevated layer that broke off the primary Raikoke plume and continued to ascend as it moved southward.

Within the context of the single-source case-study events, this could be interpreted as a mixture of smoke and sulfuric acid aerosol at these altitudes, though not necessarily at the same longitude (i.e., the spectra classified as smoke and sulfuric acid

were not necessarily part of the same airmass; please see Sect. 6.4 for discussion of potential missclassifications). What stood out in Fig. 15 was the dominance of sulfuric acid aerosol classifications at the lowermost altitudes ($\leq$21 km) and the stark transition to a predominantly smoke classification at higher altitudes (e.g., >98% at 23 km). This transition was most notable between 21 and 25 km where the slope rapidly changed (Fig. 15, panels c-g). This is the opposite behavior observed for the main peak (Fig. 14). While smoke classifications were present throughout the profile, this partitioning is what would be

expected from single-source events: the absorbing species (smoke) rose to higher altitudes while the non-absorbing species (sulfuric acid) was carried along the same altitude.

Unfortunately comparison with CALIOP was not possible for the secondary plume. Due to the sparseness of SAGE coverage in the tropics an observation that was collocated with CALIOP was not found. While Chouza et al. (2020) demonstrated that elevated layers in this latitude band were observed from both the Mauna Loa ground-based lidar and CALIOP, the CALIOP

depolarization ratio was too low to indicate smoke. The challenge in using the CALIOP depolarization ratio at these altitudes is the limited return signal. However, as explained below, it is likely that the majority of these Raikoke smoke classifications may be incorrect.

These results raise several interesting questions, all of which will be addressed below:

1. Was there smoke in the stratosphere during this time and how ubiquitous was it on a hemispherical scale?

2. If insufficient quantities of smoke were present to account for these classifications then what drove these classifications?

3. Does this interpretation fit the current scientific paradigm for converting $SO_2$ to sulfuric acid and particle growth/coagulation rates?

4. Does the Raikoke eruption fall outside the applicable bounds of this classification method?

### 6.3 Smoke present in stratosphere before and after Raikoke eruption

Though there were multiple pyroCb events in the northern hemisphere during the summer of 2019 (e.g., Siberia and western Canada per Bachmeier (2019), Johnson et al. (2021), Kloss et al. (2021), and Vaughan et al. (2021)), there were no wildfire events of similar magnitude as the 2017 Canadian wildfire or the 2020 Australian burn. However, Vaughan et al. (2021) observed stratospheric smoke prior to the Raikoke eruption, Kloss et al. (2021) suggested the northern hemisphere wildfires impacted the stratosphere, and smoke was identified in the lower stratosphere via CALIOP, ground-based and shipborne lidar

observations before and after the eruption (Ansmann et al., 2021; Ohneiser et al., 2021). To further demonstrate that smoke was in the stratosphere prior to the Raikoke eruption we evaluated SAGE and CALIOP data collected to the west of Raikoke (i.e., upwind, see Fig. 16). Unfortunately, SAGE did not begin sampling this latitude band until after the eruption, therefore we limited our analysis to the first two weeks after the eruption and to a region far enough west of Raikoke to not have been impacted by the volcanic ejecta (Kloss et al., 2021).

The CALIOP backscatter profile (Fig. 17, panel (a)) showed enhanced backscatter from 10–15 km (42$^\circ$N and 55$^\circ$N) that had significant depolarization (Fig. 17, panel (b)), indicative of stratospheric smoke. The SAGE data showed enhanced extinction at

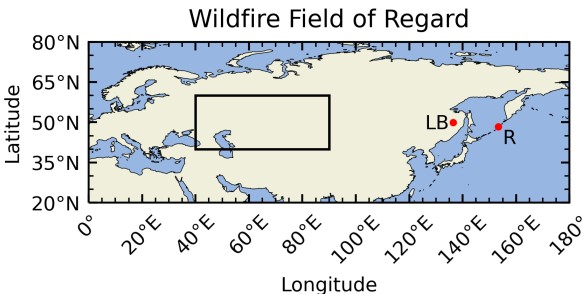

**Figure 16.** Field of regard for sampling air that has not been impacted by the Raikoke event, but may have been impacted by coincident northern hemisphere wildfires. LB and R indicate location of the Lake Bolon, Siberia fire and Raikoke, respectively.

$\approx$11 km and 19 km, with a minor enhancement at 15.5 km (Fig. 17, panel (d)). The spectral slope (Fig. 17, panel (e)) indicated the presence of smoke from $\approx$10–15 km (i.e., where slope >-0.001 nm$^{-1}$), but did not indicate smoke for the layers at 15.5 km or 19 km, in agreement with the CALIOP data. Overall, both SAGE and CALIOP indicated smoke was present between 10

and 15 km, though the backscatter and extinction profiles showed different features (e.g., SAGE saw a layer at 20 km while CALIOP did not). This is not unexpected because of the distance between the 2 measurement locations (482 km). Regardless of these minor differences, both instruments clearly showed enhanced smoke above the tropopause over a region that was yet to be impacted by the Raikoke eruption (i.e., at this time the Raikoke plume was still located to the east of the North Pacific Ocean and Canada as shown in Fig. 12, Kloss et al. (2021), and Vaughan et al. (2021)). While the SAGE sampling schedule

did not allow us to evaluate profiles collected within this region prior to this time, this smoke layer was persistent over this region, in both the SAGE and CALIOP records, throughout the first $\approx$2 weeks after the eruption (data not shown). This is in agreement with previous studies that reported observing stratospheric smoke between 8 and 13 km between 50°N and 85°N during this time period (Ansmann et al., 2021; Johnson et al., 2021; Ohneiser et al., 2021; Osborne et al., 2021; Vaughan et al., 2021).

Another opportunity for identifying air masses that were impacted by biomass burning (BB) events is to look for enhanced BB chemical tracers such as carbon monoxide (CO), and hydrogen cyanide (HCN), both of which are products within the ACE-FTS data (Boone et al., 2020). If a layer of enhanced BB markers is identified within an ACE-FTS profile, then individual ACE-FTS spectra can be analyzed to look for smoke-specific signatures (e.g., the carbonyl (C=O), hydroxyl (O−H), and C−H stretch features near 1,740 cm$^{-1}$, 3,250 cm$^{-1}$, and 1,740 cm$^{-1}$, respectively; per Boone et al. (2020)) throughout the entire

profile. To this end, we identified profiles within the ACE-FTS record that showed enhanced BB tracers, a subset of which are presented in Fig. 18. Figure 18 shows ACE-FTS profiles of CO, HCN, and SO$_2$ that were collected within 2 months of the eruption between 24.5°N and 51.9°N. The climatological profiles were created by computing the average and standard deviation of the ACE-FTS products collected within the specified latitude ($\pm$2.5°) and within the current season (i.e., July, August, September) using the entire ACE-FTS record. The open circles in Fig. 18 indicate the highest altitude at which smoke

was identified using the carbonyl-stretch method as provided by the ACE-FTS team (C. Boone, personal communication,

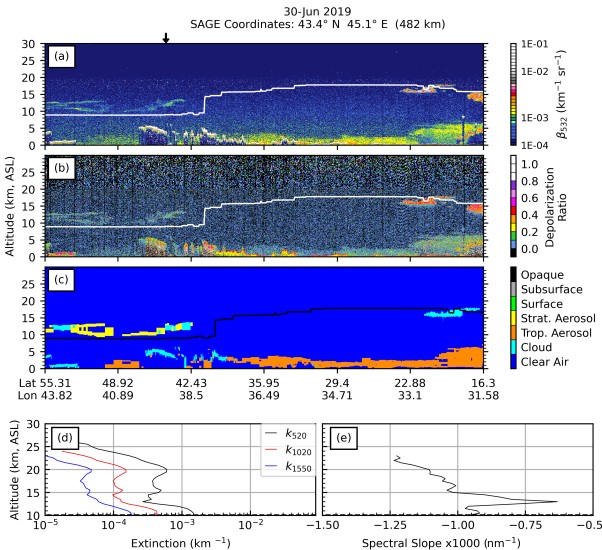

**Figure 17.** Smoke identified within an air mass that has yet to be impacted by the Raikoke eruption, but was sampled within 8 days of the Raikoke eruption. The CALIOP granule used to generate this image is CAL_LID_L1-Standard-V4-10.2019-06-29T23-42-21ZN.

2022). Finally, the $SO_2$ profiles were included to observe when the atmosphere was impacted by both wildfire and volcanic activity.

While Figs. 17 & 18 demonstrate that smoke was in the stratosphere before and after the Raikoke eruption, we were unable to find support for enough smoke to account for the quantity of smoke classifications in the SAGE data. The 2019 pyroCb activity was too limited to yield sufficient smoke to reproduce this classification, and the observational evidence from multiple instruments (e.g., CALIOP, ACE-FTS) did not support such a widespread impact from BB events. Therefore, we conclude that these smoke classification were incorrect and present support for this conclusion below.

## 6.4 Potential misclassification of Raikoke's sulfuric acid layers

Up to now the data collected during the Raikoke time period has been evaluated within the context of the single-source case study events. This was done with the perspective that the Raikoke eruption, which injected significantly more $SO_2$ than the Ambae and Ulawun eruptions, would yield sulfuric acid particles of comparable size as Ambae and Ulawun. However, if we look at this event from another perspective (i.e., as a much larger eruption, that may have created larger sulfuric acid particles than Ambae and Ulawun), then there may be an alternative interpretation for these smoke classifications.

### 6.4.1 Evidence for misclassification based on ACE-FTS comparison

While Figs. 17 & 18 demonstrate that smoke was in the stratosphere in the months following the Raikoke eruption and was within the latitude range of this study, it does not provide confirmation for all of our smoke classifications. Indeed, we were

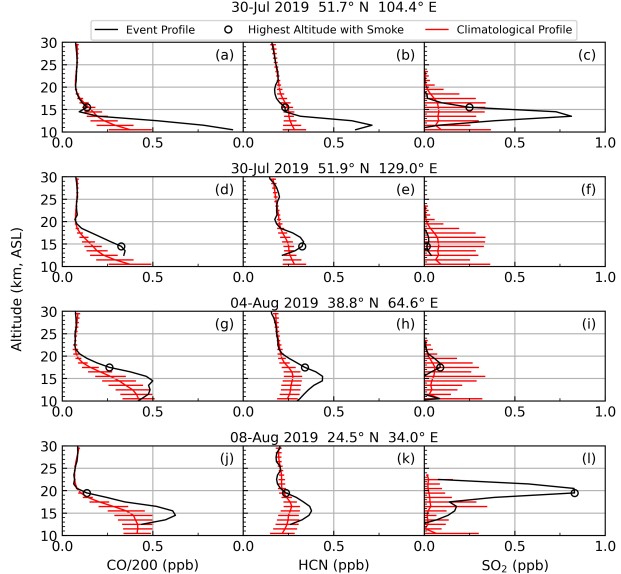

**Figure 18.** ACE-FTS profiles of CO (panels a, d, g, j), HCN (panels b, e, h, k), and $SO_2$ (panels c, f, i, l) collected within 2 months of the Raikoke eruption. The climatological profiles present the mean $\pm 1$ standard deviation for data collected within the specified latitude ($\pm 2.5°$) for July/August/September over the lifetime of the ACE-FTS instrument. The open circles indicate the highest altitude at which the carbonyl stretch was identified (C. Boone, personal communication, 2022).

unable to identify smoke or ash in the ACE-FTS data above $\approx 20$ km in any profile, while the SAGE-based classification method identified more than 2,000 enhanced spectra above 20 km. One shortcoming of the ACE-FTS identification method is it is not automated and requires visual inspection of profiles for enhanced BB tracers followed by visual inspection for spectral

residuals as described in Boone et al. (2020). Therefore, we recognize the possibility that additional smoke layers may be detected in the ACE-FTS data at altitudes higher than 20 km. While this is a possibility, the amount of smoke observed in the Fig. 18 profiles (based on the magnitude of the C=O), hydroxyl (O−H), and C−H residuals) was insufficient to account for the smoke identified in the SAGE data. Therefore, it is reasonable to conclude that the aerosols that were designated as smoke in Fig. 14 & 15 were misclassified large sulfuric acid particles.

As an illustration of this misclassification, there was one set of SAGE and ACE-FTS profiles (Fig. 19) that were exceptionally close in both space (distance of 32 km) and time (difference of 2 minutes) that allows for a direct comparison. Our algorithm indicated that smoke was present from 12–19 km. However, the ACE-FTS profile showed no enhancement of the BB tracers (only CO shown in Fig. 19) and there was no carbonyl-stretch residual (C. Boone, personal communication, 2022). Therefore, we conclude that these smoke classifications were incorrect and that the particles created by the Raikoke eruption were too

large to be applicable to the current classification method.

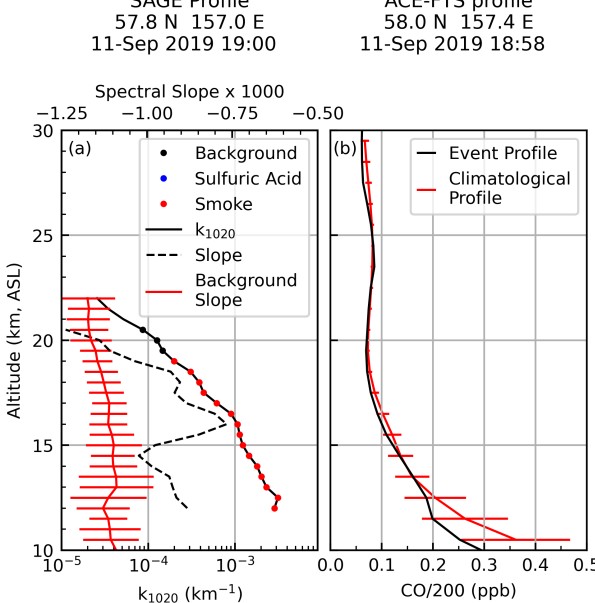

**Figure 19.** SAGE and ACE-FTS profiles collected in close proximity (both temporally and geographically). Colored dots in panel (a) indicated the SAGE-based composition classification. Panel (a) also shows the calculated spectral slope for this profile as well as the background slope values ($\tilde{X}_m \pm \mathrm{MAD}_m^*$). Panel (b) presents the ACE-FTS CO profile.

### 6.4.2 Evidence for misclassification based on size estimates

As previously discussed, large sulfuric acid particles (>200-250 nm) are expected to be indistinguishable from smoke, ash, or clouds within the context of this classification methodology (see, e.g., Fig. 1). While getting accurate particle size information from remote-sensing observations is challenging, both the ACE-FTS and SAGE instruments are capable of providing estimates

for the underlying distribution's mode radius (e.g., Heintzenberg et al. (1981); Fussen et al. (2001); Bingen et al. (2006); Wrana et al. (2021)). Mode radius estimates for the particles classified as smoke in Fig. 19 were calculated using the ACE-FTS data (C. Boone, personal communication), and the method presented in Wrana et al. (2021) for the SAGE data (F. Wrana, personal communication). A third method of inferring particle size involved finding where the slopes from Fig. 19 intersected the "Sulfuric Acid" curve in Fig. 1 (b). These estimates are summarized in Table 5.

We note the remarkable agreement between the ACE-FTS estimates and those based on Fig. 1 (panel b). These results are interesting because, based on the estimated radii and the calculated slopes of Fig. 19, neither BrC nor BC are required to intersect the "Sulfuric Acid" line of Fig. 1 (panel b). This is in agreement with the ACE-FTS composition results. On the other hand, if the particles were ≈135 nm (as indicated by the Wrana method) then a combination of BrC and BC is required to achieve the calculated slopes. While the discrepancy in the radius estimates highlights the need for ongoing work to better

understand the applicability and veracity of these particle size distribution inference techniques as well as ongoing analysis of

| Estimation Method | Radius Range (nm) | $\sigma$ Range |
|---|---|---|
| ACE-FTS | 197–230 | 1.3–1.4 |
| Wrana | 135 | 1.6–1.8 |
| Inferred from Fig. 1 | 180–220 | 1.5 |

**Table 5.** Size and distribution width estimates of the particles that were classified as smoke in Fig. 19. Estimates were made using the ACE-FTS retrieval (provided by C. Boone, personal communication), the Wrana et al. (2021) method (provided by F. Wrana, personal communication), and using information from Fig. 1 (b).

the Raikoke event to better understand the composition of stratospheric aerosol in the months subsequent to the eruption, we conclude that the smoke classifications in Fig. 19 are better described as large sulfuric acid particles. These results point toward the Raikoke particles being too large for application to the current method.

### 6.4.3 Evidence for misclassification based on time series

While the SAGE instrument did not sample the Raikoke plume immediately after the eruption, it did make observations of the Raikoke plume within 10 days of the eruption. Using the method described in Sect.4, the aerosol were classified using data from all profiles collected between June 2017 and January 2021 at 48°N ($\pm 5°$). The time series, broken into 2 km bins, is shown in Fig. 20.

       We observed a rapid increase in the spectral slope and a corresponding increase in smoke classifications immediately af-
ter the 2017 Canadian pyroCb and the atmosphere did not return to background conditions until between March/April 2018 (12–14 km, panel e) and March/April 2019 (20–22 km, panel a). We note that these smoke classifications were corroborated by correlative data (not shown) and deemed correct. Immediately prior to the Raikoke eruption there was no indication of enhanced smoke in the time series; a stunning absence if northern hemisphere pyroCbs were to have played a significant contribution to stratospheric aerosol. However, elevated sulfuric acid was detected up to 6 months prior to the eruption (potentially
transport from Ambae, per Kloss et al. (2020)). Further, the time required to return to background conditions post-Raikoke was longer than the time required to return to background after the Canadian pyroCb. This serves as an indicator of the magnitude of the Raikoke event and, if these post-Raikoke smoke classifications were correct, this would require pyroCb that injected more smoke into the stratosphere than the 2017 Canadian event. Further, this single pyroCb, or series of smaller concurrent pyroCbs, would have to have fortuitously erupted within 1–2 weeks of the Raikoke eruption. No such events were observed by
any satellite instruments, in local reports, or by commercial or private pilots. While we recognize that smoke was in the stratosphere during and after the Raikoke eruption (as demonstrated above), this was insufficient to cause this level of perturbation. Therefore, the time series data point toward a single aerosol type (sulfuric acid) and the Raikoke particles being larger than can be evaluated with the current method.

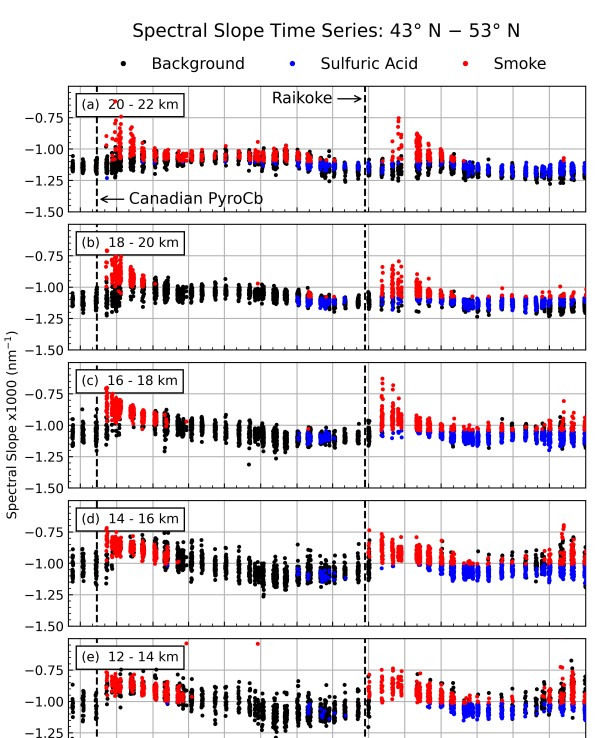

**Figure 20.** Time series of the spectral slope from June 2017 through January 2021 between $43°$N and $53°$N. The vertical dashed lines indicate the start of the 2017 Canadian pyroCb and the 2019 Raikoke eruption.

### 6.4.4 Interpreting the misclassification analyses

The question now at hand is: if the SAGE-based smoke classifications in the Raikoke event are in fact misclassifications, then why? We see several possibilities.

1. **Sulfuric acid particles were large:** The most immediate explanation would be that there were enough sulfuric acid particles to push $k_{1020}$ out of the background classification and these particles grew large enough to push the slope into the smoke classification. It is possible that due to the amount of $SO_2$ that Raikoke injected into the stratosphere 595 that the particles were substantially larger than particles from Ambae and Ulawun. While we recognize this as the most likely option, and note that the corroborative data support this conclusion, the challenge of this option is in the rate at which $SO_2$ converts to sulfuric acid (e-folding time of $\approx$30 days) and the rate at which sulfuric acid coagulates to form large particles. For this to work within the timescale of 10 days (as indicated by Fig. 20) the rate of conversion from $SO_2$ to sulfuric acid and the subsequent coagulation to yield particles with radii $>\approx$180 nm would be faster than 600 observed for previous volcanic eruptions. While previous studies observed $SO_2$ formation within days of eruption Guo

et al. (2004); Penning de Vries et al. (2014), they did not observe large particles (e.g., >180 nm) within this time period. However, Thomason et al. (2021) noted that pre-eruption aerosol loads may significantly modulate particle formation rates and Kloss et al. (2020) showed that aerosol from Ambae had migrated to Raikoke's latitude by the time of its eruption (this may be observed in the months leading up to Raikoke in Fig. 20). Therefore, it is feasible that particles from Ambae's 2018 eruption may have influenced particle growth rates within the Raikoke plume. We note this as an interesting possibility that should be studied further.

2. **Small-particle ash was present:** The 2019 Raikoke eruption was the largest eruption since the 2011 (Puyehue-Cordón Caulle) and contained up to 1.8 Tg of ash (Muser et al., 2020). While large ash particles were quickly removed from the atmosphere via sedimentation (Muser et al., 2020), it is possible for fine-mode ash (<300 nm) to remain in the stratosphere for more than 3 months (Vernier et al., 2016). Therefore, we consider the possibility that the post-Raikoke stratosphere was laden with sub-micron ash, which would increase the extinction coefficients and slope and yield a smoke classification. Further, if ash was the cause of the smoke classification then, as this ash was removed from the stratosphere, the aerosol classification would shift back toward enhanced sulfuric acid. This scenario fits the observed behavior of the time series (Fig. 20) after ≈January 2020.

While this scenario provides a reasonable explanation for the immediate smoke misclassifications and would allow ample time for $SO_2$ conversion to sulfuric acid, the challenge is that, to our knowledge, no group has reported observation of a substantive and persistent ash presence more than ≈1 month after Raikoke erupted. Further, the CALIOP depolarization ratio and VFM product should readily detect and identify stratospheric ash, but there is no indication of ash within the CALIOP record. Similarly, the ACE-FTS spectra should provide evidence of ash, should it be present. However, ash signatures are distinctly absent post-Raikoke (C. Boone, personal communication). For these reasons, we do not consider this a viable explanation.

3. **Smoke from small wildfires had a disproportionate impact:** Another possibility is that smoke was in the stratosphere, but did not make up a large fraction of the total particle count and was below the detection threshold for observing the carbonyl stretch in the ACE-FTS residuals spectra. For example, the maximum change in slope (relative to the background sulfuric acid slope) in Fig. 19 occurred at 16 km where the slope changed by 32%. Using the 70 nm background curves in Fig. 3 as a guide, we see that the amount of smoke required to shift the spectral slope by 32% ranged from 0.5% for pure BC to 0.9% for pure BrC. However, if the atmosphere was previously perturbed by a volcanic or wildfire event (i.e., as suggested by Kloss et al. (2020)) then the 150 nm curves of Fig. 3 indicate that the amount of smoke required to shift the spectral slope by 32% ranged from 6% (pure BC) to 35% (pure BrC). Because of the multiple assumptions that went into the creation of Fig. 3 we do not present these numbers as an estimate of the relative amount of smoke in the stratosphere at this time. However, what this indicates is how disproportionate the impact of smoke may be as compared to enhanced sulfuric acid aerosol as well as the importance of correctly characterizing the background statistics.

While this possibility highlights the sensitivity of the spectral slope to particle composition, it fails to account for the absence of pre-eruption smoke throughout the stratosphere and does not provide a viable pathway for explaining the shift

from smoke classifications to enhanced sulfuric acid ≈6 months after the eruption (Fig. 20). Further, smoke residuals were absent in the majority of ACE-FTS spectra analyzed for this study indicating that smoke in the stratosphere was not widespread enough to impact the majority of SAGE profiles. Finally, if smoke were present in sufficient quantities to rival the signal of the 2017 Canadian pyroCb then why was it not detected prior to the eruption of Raikoke. While we note the disproportionate impact smoke can have on the spectral slope, this option strains the limits of credulity within the current context.

While none of these scenarios provide a perfect fit to the current scientific paradigm, we conclude that the fortuitous injection of smoke immediately after the Raikoke eruption is the least likely scenario. Further, we note the lack of observational evidence for persistent ash within the applicable latitude band. Therefore, we conclude that the post-Raikoke stratospheric aerosol is best characterized as large sulfuric acid particles while admitting the possibility of an ash contribution. While this classification method performed well for smaller volcanic eruptions as well as pyroCbs, the sulfuric acid particles formed after the Raikoke eruption were too large to be applicable to this classification method.

## 7    Conclusions

We presented a method of distinguishing between sulfuric acid aerosol and smoke using the SAGE III/ISS UV/Vis/NIR extinction spectra. This methodology was evaluated in 4 case-study events (2 volcanic, 2 pyroCb) and using the CALIOP depolarization ratio and vertical feature mask. The CALIOP data were supportive of the smoke/sulfuric acid aerosol identifications. The classification of the 2 volcanic events (Ambae and Ulawun eruptions) was nearly monolithic in correctly classifying the particles as sulfuric acid (i.e., >99.5% correct classification). Identification of aerosol source for the wildfire events was more challenging in that a non-negligible fraction of the spectra were identified as sulfuric acid aerosol. However, this method still correctly classified smoke plumes more than 81% of the time.

While we cannot provide a clear definition of magnitude of event required for this method to be applicable, we can state two general cases where it is not applicable and the information provided within the introduction should allow the reader to put these events in their proper perspective. First, very small wildfire events may not inject enough smoke into the stratosphere to sufficiently change spectral slope from background conditions. Similarly, even for large events, if the SAGE sample volume only contains optically thin smoke layers (e.g., viewing through thin filaments as opposed to through the plume's centroid), this too could lead to a sulfuric acid classification. In these cases, we note that classifying these layers as sulfuric acid is not necessarily wrong since the majority of the particles within the sample volume are likely composed of sulfuric acid; this just fails to identify the presence of smoke as the signal is too small.

Second, volcanic eruptions that inject large amounts of $SO_2$ into the stratosphere (e.g., Pinatubo and the 2019 Raikoke eruptions) result in particles that are too large for this method to evaluate. Similarly, eruptions that inject large amounts of ash into the stratosphere will likewise not be applicable to this method. In both of these scenarios the particles will likely be classified as smoke instead of sulfuric acid. However, this method is applicable to all smaller events within the SAGE III/ISS record to date.

The ACE-FTS products and spectra were used to identify the presence of smoke in the stratosphere between 20°N and 60°N. Indeed, smoke was identified within these latitudes and within the stratosphere using the ACE-FTS data, but the ACE-FTS data showed no spectral evidence of smoke above ≈20 km. The ACE-FTS team estimated sulfuric acid particle radii on the order of 200 nm, which was comparable to our estimates. Therefore, we conclude that it is reasonable that the Raikoke eruption produced large sulfuric acid particles, which fall outside the applicable range of this methodology.

One aspect we did not consider within this study was the possibility of mixed composition particles. As discussed above, recent studies (Ansmann et al., 2021; Magaritz-Ronen and Raveh-Rubin, 2021; Ohneiser et al., 2021) have hypothesized that smoke that comes from non-pyroCb fires have a longer residence time in the troposphere wherein these particles undergo chemistry and adsorb water (presumably sulfuric acid as well) as they continue to loft to higher altitudes before eventually entering the stratosphere. If this is true, then these smoke particles would be significantly different from smoke particles that are injected directly into the stratosphere from pyroCb activity. These studies demonstrated that the aerosol layers they identified as smoke exhibited different spectral behavior than what has been traditionally expected for smoke. While we can neither verify nor refute the conclusions of these studies, we find this to be an interesting possibility for future investigations. If these particles were coated with water or sulfuric acid and had a carbon core, then the question must be raised as to how this would impact the spectral slope analysis as well as the interpretation of the presence, or lack thereof, of the carbonyl smoke signature in the ACE-FTS data. Indeed, this merits further evaluation.

The stratosphere has provided many interesting and challenging puzzles during the Raikoke time period. Indeed, recent publications such as Ansmann et al. (2021), Magaritz-Ronen and Raveh-Rubin (2021), and Ohneiser et al. (2021) have challenged the accepted view of how smoke enters the stratosphere, how stratospheric smoke may be identified in lidar data, as well as the interpretation of CALIOP data. While their findings lend support for stratospheric smoke during the Raikoke time period, they are contrary to what was observed in ACE-FTS and fall outside the current scientific paradigm. Therefore, at this point, the overall conclusions of the disparate analyses are incongruous and we view this as an opportunity to either better understand the interpretation of our respective data sets or to better understand a previously unrealized aspect of the atmosphere (i.e., all of the research groups are right, but the results are seemingly incongruous because they are only seeing one piece of the puzzle). Indeed, the post-Raikoke stratosphere merits further detailed investigation to clarify these disparate conclusions and elucidate the corresponding chemistry, composition, and physics of these layers.

*Data availability.* The SAGE and CALIOP data used within this study are available on NASA's Atmospheric Science Data Center (https://eosweb.larc.nasa the TropOMI data are available at the Copernicus Open Access Hub (https://scihub.copernicus.eu/), and ACE-FTS data are available from the ACE/SCISAT Database (https://databace.scisat.ca/).

*Author contributions.* TNK and LT developed the methodology, while TNK carried out the analysis, wrote the analysis code and the manuscript. RD, MK, DF, JT, and JK participated in scientific discussions and provided guidance throughout the study. JT and JK provided guidance on the use and interpretation of the CALIOP data. All authors reviewed the manuscript during the preparation process

*Competing interests.* The authors declare that they have no competing interests.

*Acknowledgements.* SAGE III/ISS is a NASA Langley managed mission funded by the NASA Science Mission Directorate within the Earth Systematic Mission Program. Enabling partners are the NASA Human Exploration and Operations Mission Directorate, International Space Station Program and the European Space Agency. SSAI personnel are supported through the STARSS III contract NNL16AA05C.

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
