# Peer review of "Identification of Smoke and Sulfuric Acid Aerosol in SAGE III/ISS Extinction Spectra"

_Atmospheric Measurement Techniques, 2021_

## Referee Comment (RC2)

**Review of Knepp et al. (AMT-D, 2021), "Identification of Smoke and Sulfuric Acid Aerosol in SAGE III/ISS Extinction Spectra Following the 2019 Raikoke Eruption"**

**Reviewer: Mike Fromm**

**Note: Material excerpted from the manuscript is colored red.**

**Note: Technical comments are shown as comments placed within the manuscript, appended.**

**Knepp et al. (hereafter K21), focus on SAGE III/ISS multi-spectral aerosol extinction data to assert the claim that pyrocumulonimbus-injected smoke makes a significant contribution to northern hemispheric stratospheric aerosol abundances in the same timeframe as the well-recognized volcanic sulfate plume generated by the Raikoke eruption of June 2019. K21 rely on multi-spectral aerosol-extinction ratio (and slopes derived therefrom) within plume measurements to discriminate between volcanic sulfates and smoke. They develop**

their thesis from SAGE III/ISS observations and a theoretical model, both showing a tangible difference in the spectral change in extinction between smoke and sulfate. They buttress their argument with cited statements about observed "large-scale pyroCb events" in spring 2019.

K21 conclude that the Raikoke sulfate plume was substantially blended with pyroCb smoke for months. Their sulfate and smoke classification method resulted in smoke detections at all altitudes between 14-25 km altitude. These findings included the extraordinary result that smoke classifications exceeded sulfate classifications on a proportional basis up through 22 km in the "Raikoke Primary" plume, and between 22-24 km in the so-called "Raikoke Secondary" plume, according to their Table 3. They hypothesize that shortwave absorption by the smoke component was an ingredient in the apparent diabatic rise of the Raikoke plume. Consequently, the reader might infer a quantifiable smoke fraction from these data insofar as the smoke is responsible for the considerable diabatic lofting of the Raikoke sulfate plume.

Such extraordinary conclusions require extraordinary evidence. Although K21 provide interesting SAGE III/ISS aerosol extinction patterns in the Raikoke timeframe and previously, their thesis and presentation are wholly unconvincing. It is demonstrable that a primary reliance on aerosol-extinction spectra in the visible-to-near IR realm for inferring aerosol composition is ill advised. In short, SAGE-type aerosol extinction ratio is under constrained for such purposes. This is made obvious by invoking Thomason et al. (2021), (hereafter T21) who blended SAGE II and SAGE III/ISS extinction-ratio data in an exploration of 10 stratospheric volcanic sulfate plumes. Not only did T21 treat the Raikoke-season aerosol as an exclusively sulfate composition, they showed that sulfate-plume extinction spectra occupied a range of values enveloping K21's smoke characteristics. T21's summary Figure 8 showed that volcanically perturbed sulfate .5-1.0 micron extinction ratios ranged by a factor of 3-5 among the 10 plumes they analyzed. These 10 events displayed an equally likely positive and negative transition from background to perturbed extinction ratio.  Taking T21 and K21 together, it became evident to me that SAGE smokeplume visible-NIR extinction spectra fall within the wide range of observed sulfate-plume extinction spectra. I can only conclude that I misunderstood the arguments/findings of K21 (and the earlier T21) or that the premise is fundamentally weak. If the first conclusion applies, I would recommend a thorough clarification of K21's fundamentals and reconciliation with T21. If the second conclusion applies, this work does not merit publication.

K21 contains additional major weaknesses that would need to be addressed for this theme to merit publication. They are elaborated on below.

**Major Issues**

**Figure 1.** This sets the stage by displaying 4 stratospheric aerosol layers that K21 attribute to Raikoke. If the date and latitude/longitude coordinates of the 4 profiles are accurate, 3 of the 4 layers are not Raikoke material. K21, elsewhere in the manuscript, correctly state that by the time of those profiles the

Raikoke plume had not advanced to 3 of the 4 positions (eastern Atlantic to Europe). Only the profile in Figure 1d is in a location (Canada) consistent with the spreading Raikoke plume. Panels a-c show smoke layers connectable to a Canada pyroCb in mid-June 2019. This is a major concern only in that it demonstrates an internal inconsistency, introduces the reader to 3 misleading profiles, and combines sulfate and smoke profiles under a single sulfate banner. If this figure is to be retained, K21 are encouraged to re-select profiles for display and rigorously qualify them based on convincing complementary data (such as maps of Raikoke $SO_2$).

Line 95. "A unique combination of volcanic and pyroCb events occurred in 2019 when the eruption of Raikoke was preceded by pyroCb events in Canada and Russia." K21's characterization is inaccurate. There was nothing unique about 2019. PyroCbs occur every year. They were also abundant/notable in 1991, 2008, 2009, and 2011, when Pinatubo, Kasatochi, Sarychev Peak, and Nabro created massive sulfate plumes. Indeed Fromm et al. (BAMS, 2010) pointed out a significant pyroCb injection in summer 1991 that was sampled by SAGE II, and may have been a contributor to the "new mode" of

aerosol particle sizes suggested by SAGE II extinction spectra that year (Thomason, 1992). K21 are correct in recognizing that the co-presence of smoke and sulfate presents a measurement-interpretation challenge—in 2019 and other years. The realization that smoke has been on multiple occasions a non-negligible neighbor of stratospheric sulfates, and T21's illustration of broad ranges in visible-NIR extinction spectra in volcanic plumes (and volcano-pyroCb blends), heightens the improbability that SAGE-like extinction spectra alone are sufficient for particle-type attribution. For these reasons, K21 are advised to invoke complementary satellite data toward a more convincing discernment between volcanic and pyroconvectively sourced plume compositions based on SAGE data.

Section 4. In this section K21 present a theoretical approach to understanding Vis-NIR extinction spectra for absorbing and scattering media. Smoke and sulfate are distinguished solely on the basis of the spectral variation of refractive index. This is interesting and informative. However, K21 state explicitly that their theoretical construct is "…in no way intended to be representative of actual conditions." This can be seen as a major weakness in that Figure 3b shows that an effective radius disparity between brown carbon and

sulfate can be as small as ~10 nm for an equal slope of -1.5, ~25 nm for -1.0, and 60 nm for -0.5. One can infer from the work of T21 (their Fig. 8) that the range of effective radius for a variety of sulfate plumes is quite large, perhaps exceeding 60 nm. Given the paucity of information on stratospheric smoke-plume and sulfate-plume mode radius, and the reasonable expectation of systematic differences, it seems absorption systematics might be potentially inconsequential in the case of certain "actual conditions." In short, the theoretical experiment offers only one real-life particle-population systematic when others (to wit, particle size) will likely muddle plume distinction. If this paper is to rely strongly on theoretical underpinnings, the simulations must be multi-faceted.

Section 7.1. K21 state "…there were two pyroCb events in the northern hemisphere during the summer of 2019…" and cite Kloss et al. (2021), Vaughan et al. (2021) and Bachmeier (2019). Neither Kloss et al. nor Vaughan et al. provide concrete details of any pyroCb event; neither goes much farther than to claim that pyroCbs occurred in Canada and Russia. No information is provided on the massiveness of these injections, a crucial element. The Bachmeier citation refers to a blog post about a pyroCb in eastern Siberia on 30 April 2019.

It is known, and can be gleaned from the Bachmeier post, that that pyroCb is highly unlikely to have been a major contributor to the stratospheric aerosol burden. Moreover, it occurred seven weeks prior to the Raikoke eruption. If this was considered a candidate for all the smoke K21 detected in 2019, it is incumbent on them to investigate that event much more deeply and quantitatively. The same goes for the Canada pyroCb event. K21 rightfully acknowledge that the pyroCb action in 2019 did not match noteworthy pyroCb events such as the ones in British Columbia, 2017, and Australia in 2019/20. These, and a few others, were quantitatively massive, long lasting, and involved stratospheric plume lofting to the altitudes of the Raikoke plume in 2019. Presumably, any pyroCb that would have made a suitable contribution to stratospheric smoke in 2019 would be easy to identify. Relying exclusively on the vague information and citations provided herein is insufficient to buttress the extraordinary claims made by K21. In truth, there were in excess of 30 boreal pyroCbs in 2019. Some occurred prior to the Raikoke eruption, and several occurred thereafter. At least three were demonstrably large enough to create traceable intercontinentally transported plumes. Evidence of one of these plumes was inadvertently demonstrated by K21 in Figure 1.

Hence, as in the Pinatubo summer of 1991, it is likely that stratospheric smoke was competing with Raikoke sulfates. But as argued above, this implies an obligation to apply much greater rigor in composition determination, necessarily involving several additional complementary data sets. SAGE data alone are insufficient. K21 are encouraged to either challenge that assessment or radically bolster their data analysis.

An example of the above suggestion was demonstrated by Cameron et al. (2021), who combined profile retrievals of $SO_2$ and aerosol extinction in an examination of several stratospheric volcanic plumes, one of which was Raikoke. By exploiting coincident volcanic-gas and aerosol profiles, they presented a first-order confirmation of sulfate particulate matter. In the case of Raikoke, Cameron et al. demonstrated close association of $SO_2$ and aerosol enhancements in what K21 consider the "Raikoke Primary" and "Raikoke Secondary" plumes. This of course does not rule out some minor influence of biomass-burning-generated aerosol, but it clearly shows a picture of volcanic material over the altitude/zonal/temporal range examined by K21. Presumably, if K21's assertion of smoke-dominant presence is verifiable, complementary data embodying biomass burning signatures such as

carbon monoxide would aid in identifying a sulfate/smoke blend. To make their case, K21 are encouraged to leverage data sets such as ACE-FTS and Aura Microwave Limb Sounder (e.g. https://acp.copernicus.org/articles/21/16645/2021/ in addition to CALIOP (exploiting its depolarization ratio data item).

Table 3. Canada. The breakdown between sulfate and smoke shows that sulfates represent more than 12% of stratospheric aerosol enhancements at all altitudes, and dominate at 23-25 km. Whereas boreal pyroCbs occur every year, extratropical volcanic events of VEI=4 do not. To my knowledge, there was no 2017 boreal volcanic eruption.  K21 do not cite any evidence of such an eruption. Thus, what is the rationale for assigning such an overwhelming number of SAGE observations to sulfate, especially at altitudes >22 km? Unless there is to be a claim of an unpublished, suspected volcanic eruption that year, it does not seem logical to categorize any SAGE measurements as sulfate. More logically, these are an indication of smoke that overlaps into sulfate extinction-spectra space, or simply uncertain. Please justify classifying any of the 2017 aerosol enhancements as sulfate.

**Section 7.1, Lines 367-369**. Indeed, it is true that there were no 2019 pyroCbs in the class of the 2017 Canada and 2019/20 Australian events. Then how to explain smoke rising to 25 km and lasting 7 months? Except for citing Kloss, Vaughan, and Bachmeier, no attempt is made to assess the magnitude of the 2019 pyroCb plumes to determine if they had the ingredients to generate such a lasting and self-lofting plume. It seems as if the authors are satisfied that the information presented in the cited works makes it self-evident. Either the 2019 pyroCbs had sufficient heft to exert the extraordinary impact reported by k21 or they didn't, in which case a novel interplay between sulfates and smoke occurred. K21 should undertake a more rigorous 2019 pyroCb survey and, depending on their finding, offer a cogent explanation for the processes that support their high-altitude, persistent smoke+sulfate anomaly.

**Line 436, 437**. "Unfortunately comparison with CALIOP was not possible for the secondary plume." K21 have missed a strategic opportunity to fully exploit CALIOP to help inform the sparser SAGE data at low latitudes. SAGE coincidences are not necessary determine the likelihood of smoke by way of CALIOP data. CALIOP depolarization ratio data are abundant from low to high

latitude and throughout the life of the Secondary plume. It would be straightforward and advisable to analyze depolarization ratio for low- and high-latitude stratospheric layers to see if there is support for smoke.

 Minor Concerns

Introduction, starting at Line 46. The next two paragraphs are interesting and well composed. But how relevant is this background to the issue at hand? I encourage k21 to prune the material here to improve the focus on the volcanoes in the satellite era.

Line124-126. What is the benefit of interpolating to 520 nm versus just adopting either 455 or 755 nm channels for analysis? I.e. what is special about 520 nm?

Line 137-138. A citation is needed for the range of injection altitudes, especially "19 km."

Line 138; callout of Fig. 1. Please see my previous comment about Fig. 1 and modify this sentence accordingly.

Line 288-289. "Both depolarization ratio and the VFM were used herein to corroborate the identification of sulfuric acid aerosol and smoke within the SAGE data." That was apparently not done for Fig. 1. CALIOP near coincidences are available and show the requisite depol. ratio for smoke. When reworking Fig. 1 and the attendant discussion, please include CALIOP coincidences.

Section 4. This is where K21 introduce the idea of a calculated slope as an alternative to extinction ratio. But I could not find where they precisely defined how slope was calculated. This would be essential for readers who would like to replicate K21's method and results. Please elaborate on the slope calculation.

Line 434, 435. The physical process described here is confusing and unclear. Smoke was shed from what? What is the cited precedent for smoke acquiring sulfate?

References

Cameron, W., P. Bernath, C. Boone, Sulfur dioxide from the atmospheric chemistry experiment (ACE) satellite, Journal of Quantitative Spectroscopy and Radiative Transfer, Volume 258, 2021, 107341, https://doi.org/10.1016/j.jqsrt.2020.107341.

[revised manuscript text omitted]

---

## Author Comment (AC1)

We would like to thank the first anonymous reviewer (AR1) for providing feedback on this manuscript. Our responses are provided below (red) to AR1's comments (black).

Title: It is surprising that the authors put the focus of their paper on the Raikoke eruption, since this event is only one of five cases under investigation and the most difficult case with the less successful results due to the complexity of the situation. They could consider changing it.

We thank the reviewer for this suggestion and we agree that a change in title may have been applicable to the original manuscript. The 4 case study events were used to test the classification scheme, which was then applied to the Raikoke event to evaluate performance during a mixed event. We believe that, with the added information present in this revised manuscript, the focus of the paper has been shifted even more toward the Raikoke event. Therefore, we believe that the current title is appropriate.

L. 49, p.2: The notation "VEI-7-8" is unclear. Please clarify. Changed

L. 118, p.4: Does the removal of all events with errors higher than 20% have a significant effect on specific time/latitude ranges by decreasing the amount of data down to a very small number of events specifically in these time/latitude intervals ? If it is the case, could this possibly induce some kind of bias in the results shown later in the paper ?

This is an insightful comment and we thank the reviewer for raising it. No, this has no impact on the date/time/latitude ranges. Rather, this just removes single data points that exceed this cutoff value. This will not result in a date/time/latitude related bias.

L. 133-135, p.5: The users are probably using Level 2 CALIPSO data in the version 4.2. On the contrary, the Level 3 data are monthly averaged. This important difference between the data level as well as the time duration used for the averaging should be specified.

The processing levels (L1 for depolarization ratio, L2 for vertical feature mask, and L3 for ASR) as well as temporal average time for the L3 product were added to the text.

L. 142, p.5: It would be useful to define the concept of "attenuated scattering ratio" or to refer to some paper where it is defined.

A brief definition for attenuated scattering ratio is now provided as is a reference to how this is used in the CALIOP algorithm.

L. 149, p.5: Do the authors mean "as early as September"? It is hard to see any trace of any secundary plume in figure 2(c) and the indication "secundary plume" is moreover shown in Figure 2(d).

No, the secondary plume is just starting to be seen in August. This is clarified in the text.

L. 185, p.7: The authors should justify their statement that BrC is more likely to be present in the stratosphere than BC, or provide a reference in this sense.

This is incorrect. Biomass burning produces a mixture of black and brown carbon with the ratio between the two dependent on burn conditions. The statement has been corrected with additional references included.

L. 187-189, p.8 and Figure 3, p.9: For the sake of clarity, the same notation notation ("1.", "2." or "(a)", "(b)") should be used here and in Figure 3. Also, the text and figure captions should be clarified. In the text: Is this simulation made for the 3 situations (sufate, BC and BrC)? In the caption: What is the reference used for the normalization? In figure (a), what is the mode radius? In figure (b), what is the reference wavelength for the calculation of the spectral slope?

This is an important clarification so we thank the reviewer for pointing this out. The text was updated to explicitly reference each panel as well as to explicitly state that all 3 compositions (sulfuric acid, BrC, BC) were simulated. The mode radius for Fig. 3, panel (a) was stated on line 187 of the original manuscript, but is now included in the figure caption for clarity. The method of normalizing each curve is now stated in the caption. Finally, there is no reference wavelength used in the linear regression. This is simply fitting a line to the wavelengths (independent variable) and extinction coefficients (dependent), prior to normalization.

Figure 3, p.3: It might look surprising, also with respect to the data in Table 1, that the sulfuric and BrC behave quite similarly (although scattering dominates for the first one, and absorption for the second one) while the BC curve have a significantly different behaviour? Do the authors confirm that there is no confusion between some curves?

Yes, we confirmed that the lines are labeled correctly and we updated this figure.

L. 204, p.8 - L.207, p. 9: If the attribution of the considered species in the 3 curves is correct, in view of the large uncertainty on/variability of the mode radius and taking into account the fact that sulfuric acid droplets have no reason to have similar sizes to BC particles, one has to imagine a large uncertainty around each curve, and the difference between sulfuric acid and BrC is unlikely to be really detectable using this method. This might be an additional reason in the presence of "false positives" in both volcanic eruption and wildfire cases.

There has been much confusion over the interpretation and utility of Figure 3 so we thank the reviewer for raising this issue and allowing us the opportunity to correct this. As stated in the original manuscript, there are many assumptions that go into creating this figure; therefore, this figure should not be considered representative of actual atmospheric conditions during any of the events presented herein. Rather, this figure presents a *very generalized guide* for how particles of differing composition may change our measured extinction spectrum. Using this figure we developed the hypothesis that we *might* be able to distinguish between smoke/sulfate using the slope method. This figure does not *prove* that this is possible; rather, the case study events speak to this.

The 2 smoke curves show a range of potential values that are dependent on the composition (or degree of "complete" combustion) of the smoke. The actual refractive index for smoke is highly variable as shown be Liu et al. 2015, and the refractive indices we chose provide a reasonable representation of the BrC RI boundaries in Liu et al. 2015's Fig. 4. As stated above, wildfire

burns result in a mixture of BrC and BC being released into the atmosphere and the BC/BrC ratio will be highly variable depending on burn conditions. Further, the composition of BrC determines its spectral properties (i.e., refractive index), which results in a wide range of possible refractive index values (as now discussed in the revised manuscript). Of course, this is all complicated by the lack of in situ measurements of stratospheric smoke. Indeed, it would seem that there is a great measurement and modeling opportunity here that should be seized, but is outside the scope of this manuscript.

Finally, the reviewer is correct that smoke and sulfuric acid particles are not expected to have the same sizes. This is especially true when considering background sulfuric acid aerosol (radii typically  $\approx$ 70-80 nm) and smoke (radii typically  $\approx$ 150-200 nm). Per Figure 3 of the manuscript, if there is 10% BC in 120 nm smoke particles then there really is no way to confuse these particles with background sulfuric acid particles. We updated this section to make these points clear to the reader as well as added a new section to discuss potential misclassifications (§§5.2 and 7.4.

Caption Figure 4, p. 11: Please repeat in the caption the relevant information provided in the figures. Character size is quite small in the figure and the information mentioned in it is difficult to read.

Font size within the figure was increased and the caption was updated to indicate extinction ratios are plotted as a function of 1 um extinction.

L. 214-216, p.9: The authors' argument is not clear. In the four cases, the light blue points corresponding to the extinction ratio with the 1020-nm channel provides flat curves, in both volcanic (with dominating sulfuric acid) and wildfire (with dominating carbonaceous aerosols) cases. Their composition is thus quite different from the background in at least one case! On the other hand, extinction ratio values change quite strongly with the 1020-extinction coefficient in the case of the 520:1550 ratio, for both volcanic and wildfire cases. What do they mean by this sentence?

We apologize, but we do not understand the issue pointed out by AR1. The original manuscript stated that for volcanic events the extinction ratios remained either unchanged or slightly larger (including for the 520:1550 ratio) while the fire events showed extinction ratios merging to similar, smaller, values. This seems to be in agreement with the reviewer's interpretation of the figure so we see no change to be made here.

L. 223-227, p.10: The fact that a Pinatubo-like eruption cannot be assessed by the present method is not related to a difference of process (in all cases, SO2 has to be converted in sulfuric acid using the available water vapour and within some characteristic formation time), but to the size of the resulting particles: if the resulting particles are very large, the spectral dependence is flat and the extinction ratio is close to 1; if the resulting particles are not large with respect to the wavelength, a varying spectral dependence is found. Therefore, the explanation provided in L. 223-224 seems not the right one. Also, the role of ashes is not taken into account in the present discussion.

We apologize, but it seems that the reviewer's comment is in agreement with what is already in the text (i.e., large eruptions like Pinatubo inject a lot of SO2 into the stratosphere, which gets converted to sulfuric acid that goes on to coagulate into large particles that yield a spectrally flat extinction spectrum).

We appreciate the reviewer pointing out the role ash may play in the spectral slope. Ash is now mentioned in this section explicitly as well as other appropriate places within the manuscript.

L. 240, 245-256, p.10: In view of the case illustrated in Figure 3, the linear regression is much more reliable if you don't consider the 1550-nm channel. Why do the authors conserve this 1550-nm channel? Starting again from the case of Figure 3, the value of the slope is likely to be very similar in the sulfuric acid and BrC cases.

This may be true depending on the refractive index and PSD parameters. Therefore, how "reliable" the regression is depends on these parameters as well. That said, all regressions for all case study events were conducted in the same way. Regarding the comment on Figure 3 in general, please see the updated discussion of this figure within the manuscript as well as our response to the reviewer's previous comment on the same topic.

L. 268-272, p.12: The authors should explain or show on z figure why the slope is more negative / flatter than the background slope for sulfuric acid / smoke.

A separate figure to explain this is not needed. First, we cannot say for certain why this happens. However, using Fig. 3 as a general guide as well as applying a general understanding of particle formation it is reasonable that the decrease in slope (i.e., more negative) is because of small particle formation; indeed this is the expected behavior for particle formation. As a general guide, Fig. 3 shows that, regardless of composition, smaller particles force the spectral slope to more negative values. This comment is now included in the manuscript.

L. 286, p.12: I suggest that the authors add the corresponding value of the depolarization ratio after "do not depolarize".

We appreciate the reviewer pointing out that this needs quantified. The following text was added to the manuscript: "This feature can be used to separate stratospheric smoke from the volcanic sulfate particles which are spherical and have only low particulate depolarization ratio (i 0.1). Prata et al. (2017) found mean particulate depolarization ratios of 0.09 and 0.05 for the sulfates from Kasatochi and Sarychev volcanoes."

Caption Table 3: The authors should specify what they mean by "Raikoke Primary" and "Raikoke Secondary", or refer to the explanation given in Section 7.2.

A reference to section 7.2 is now in the caption and an explanation is provided within the text.

L. 301-302, p.14: Where are these number coming from ? In Table 3, the fraction of misclassified events reaches a maximum of 62% at 24 km height for Ambae and 100% at 15 km height for Ulawun. Please clarify.

The original statement in the paper is correct. The reviewer is correct about the 65% and 100% misclassification for Ambae and Ulawun at 24 km and 15 km, respectively. However, the

total number of spectra identified as "enhanced" were small for these altitude/events (8 and 1, respectively); therefore, these values, while large in a relative sense, do not carry significant weight into the overall statistic.

Caption Figure 7: Please complete the caption and describe all panels to make the figure self-explanatory.

The caption was updated to contain a description of all panels within the figure.

L. 352, p.17: Where is the estimate ">81%" coming from? From Table 3, the fraction of identified smoke events is >60% if all altitudes considered, and >86% up to 24 km height.

The reviewer only considered 1 wildfire event within this comment. The original statement in the original manuscript was in regard to the overall performance for the wildfire case study events as a whole and this is indicated in the text. The sentence remains unchanged.

Figure 13: I suggest that the authors use another colour for the indication "LB" and "R", which are poorly visible.

The marker colors were changed to red and made larger. The "LB" and "R" labels were poorly visible because they sat atop other lines within the figure. The labels were moved and are now easily readable. We thank the reviewer for a suggestion that improves this figure.

L. 404, p.21: Large particle have to grow from condensation nuclei to large particle by all successive microphysical processes (condensation, nucleation, coagulation). They are thus likely to need several weeks (up to one month) to become large particles. Per se, they are expected to be short-lived, but to appear later. The case shown in Figure 14 was measured on 30 June 2019, about one week after the eruption. Hence, isn't it likely that these particles rather concern ash?

This figure shows data collected over the field of regard displayed in Fig. 13. This region was not impacted by the Raikoke eruption by this date, so this is a demonstration of the presence of smoke in the stratosphere. No change to the manuscript is required for this comment.

L. 420, p.22: Do the authors mean: "either a mixture of sulfuric acid and ash or smoke"?

Identification of ash is outside the scope of this study. Therefore, to this point we have only considered 2 possibilities: smoke or sulfuric acid (or some combination of the 2). However, the point raised by the reviewer is good and we now mention ash here as well.

L. 422-423, p.22: The authors try to distinguish sulfuric acid and smoke, but do not discuss the distinction between BC from BrC, although their respective spectral behaviours illustrated in Figure 3 look quite different. Actually, in view of the relative similarity between the cases of sulfuric acid and BrC, wouldn't it provide a plausible explanation for many "false positive" cases in all cases where wildfires take place (Australian and Canadian pyroCb and Raikoke)? It is noticeable that all these cases show a significant amount of "false positive" (see Figure 9, 10, 15, and 16) while both purely volcanic cases show only very few ones (see Figure 5-6). This is an important detail and we thank the reviewer for raising this concern. The chance for misclassifications was briefly mentioned in the original manuscript but has been expanded in this revised version. Indeed, BrC and sulfuric acid in Fig. 3 are quite similar. However, the updated discussion of this figure should make the uncertainty in defining the difference between BrC and BC more clear. In short, we cannot differentiate between BrC and BC but differentiating between smoke that has even 10% BC in it and background sulfuric acid aerosol should be readily achievable.

L. 425, p.22: Citing altitudes of 19 and 20 km could be even more convincing. We appreciate the reviewer's suggestion and agree. This change has been implemented.

L. 453, p.24: The statement is different here from above in the text (L. 199-202, p.8). The authors should replace "there is a chance for" by "the result is most likely to be", or just repeat that the method is not applicable in this case.

We agree with this comment. The pertinent sentence now reads: "Second, during large-scale volcanic eruptions (e.g., Pinatubo with VEI 6), the probability for misclassifying the large sulfuric acid particles, or ash, as smoke is high."

L. 58, p.3: Duplicated "has". Corrected

L. 130, p.5: "which" should probably be removed. Corrected

Caption Figure 1: I suggest that the authors reproduce the time and geolocation of the four events in the caption for the safe of readability.

This figure has been significantly updated from the original document. The caption was updated as the reviewer requested.

---

## Author Comment (AC2)

We would like to thank the second reviewer (R2, Mike Fromm) for providing feedback on this manuscript. Our responses are provided below (red) to Dr. Fromm's comments (black).

Such extraordinary conclusions require extraordinary evidence. Although K21 provide interesting SAGE III/ISS aerosol extinction patterns in the Raikoke timeframe and previously, their thesis and presentation are wholly unconvincing. It is demonstrable that a primary reliance on aerosol-extinction spectra in the visible-to-near IR realm for inferring aerosol composition is ill advised. In short, SAGE-type aerosol extinction ratio is under constrained for such purposes. This is made obvious by invoking Thomason et al. (2021), (hereafter T21) who blended SAGE II and SAGE III/ISS extinction-ratio data in an exploration of 10 stratospheric volcanic sulfate plumes. Not only did T21 treat the Raikoke-season aerosol as an exclusively sulfate composition, they showed that sulfate-plume extinction spectra occupied a range of values enveloping K21's smoke characteristics. T21's summary Figure 8 showed that volcanically perturbed sulfate .5-1.0 micron extinction ratios ranged by a factor of 3-5 among the 10 plumes they analyzed. These 10 events displayed an equally likely positive and negative transition from background to perturbed extinction ratio. Taking T21 and K21 together, it became evident to me that SAGE smoke- plume visible-NIR extinction spectra fall within the wide range of observed sulfate-plume extinction spectra. I can only conclude that I misunderstood the arguments/findings of K21 (and the earlier T21) or that the premise is fundamentally weak. If the first conclusion applies, I would recommend a thorough clarification of K21's fundamentals and reconciliation with T21. If the second conclusion applies, this work does not merit publication.

We thank Dr. Fromm for this thorough comment. However, we believe there is a misunderstanding here. First, the role of T21 was not to identify the composition of stratospheric aerosol. Rather, T21 operated under the assumption that sulfuric acid aerosol was the only aerosol source and they presented a particle nucleation and growth hypothesis that was supported by a very simple simulation. The findings of T21 have no bearing on composition, and the key results can be modified to accommodate differing aerosol compositions.

Per Dr. Fromm's comment that "These 10 events displayed an equally likely positive and negative transition from background to perturbed extinction ratio"; this is roughly true. However, this is expected behavior and is in agreement with the premise of the current manuscript (K21). The original manuscript had statements that this analysis will fail for large and ash-laden eruptions; the very type of eruptions that yielded the decreasing extinction ratios of T21. We believe that this was, perhaps, not clearly communicated in our manuscript, so we have reiterated these limitations throughout as well as now discuss the limitations with interpreting the Raikoke secondary plume.

T21 and K21 are not directly comparable because they consider different time periods and do not share a common goal or methodology (e.g., one used extinction ratios the other uses spectral slopes (the 2 are inversely related), one seeks to infer composition the other aims to understand aerosol physics immediately after an eruption). T21 focused on the time surrounding the background period and *maximum extinction coefficient*, while the K21 analysis used a wider time frame. This is important because, for example, while Kelut and Ruang showed a decreased extinction ratio when extinction was maximized (Fig. 7 of T21), the ratio increased to abovebackground levels shortly thereafter. Raikoke looks like it's about to follow suit, but the time series in T21 was cut off too soon. Therefore, while Kelut and Ruang showed decreased extinction ratio early on, they reversed course to look more sulfate-ish per the K21 classification criteria. This leaves the Pinatubo event, which we cannot address. We do not see the 2 studies as being contradictory to each other. While the two are quite different in methodology and purpose, we see the two studies as complementary to each other.

Figure 1. This sets the stage by displaying 4 stratospheric aerosol layers that K21 attribute to Raikoke. If the date and latitude/longitude coordinates of the 4 profiles are accurate, 3 of the 4 layers are not Raikoke material. K21, elsewhere in the manuscript, correctly state that by the time of those profiles the Raikoke plume had not advanced to 3 of the 4 positions (eastern Atlantic to Europe). Only the profile in Figure 1d is in a location (Canada) consistent with the spreading Raikoke plume. Panels a-c show smoke layers connectable to a Canada pyroCb in mid-June 2019. This is a major concern only in that it demonstrates an internal inconsistency, introduces the reader to 3 misleading profiles, and combines sulfate and smoke profiles under a single sulfate banner. If this figure is to be retained, K21 are encouraged to re-select profiles for display and rigorously qualify them based on convincing complementary data (such as maps of Raikoke SO2).

We thank Dr. Fromm this pointing this out. This figure is now updated to include SAGE extinction coefficient profiles for a single event, a CALIOP backscatter profile, as well as TropOMI  $SO_2$  map.

Line 95. "A unique combination of volcanic and pyroCb events occurred in 2019 when the eruption of Raikoke was preceded by pyroCb events in Canada and Russia." K21's characterization is inaccurate. There was nothing unique about 2019. PyroCbs occur every year. They were also abundant/notable in 1991, 2008, 2009, and 2011, when Pinatubo, Kasatochi, Sarychev Peak, and Nabro created massive sulfate plumes. Indeed Fromm et al. (BAMS, 2010) pointed out a significant pyroCb injection in summer 1991 that was sampled by SAGE II, and may have been a contributor to the "new mode" of aerosol particle sizes suggested by SAGE II extinction spectra that year (Thomason, 1992). K21 are correct in recognizing that the co-presence of smoke and sulfate presents a measurement-interpretation challenge—in 2019 and other years. The realization that smoke has been on multiple occasions a non-negligible neighbor of stratospheric sulfates, and T21's illustration of broad ranges in visible-NIR extinction spectra in volcanic plumes (and volcano-pyroCb blends), heightens the improbability that SAGE-like extinction spectra alone are sufficient for particle-type attribution. For these reasons, K21 are advised to invoke complementary satellite data toward a more convincing discernment between volcanic and pyroconvectively sourced plume compositions based on SAGE data.

This is a good point and we appreciate Dr. Fromm pointing this out. The wording was updated to remove the claim of "uniqueness". The four case-study events presented in this manuscript demonstrate that this methodology is capable of distinguishing between smoke and sulfuric acid aerosol when sampling a single-sourced plume. We realize this is not without limitations and we have expanded the discussion of potential misclassifications. However, Dr. Fromm's greater point here is that distinguishing between smoke and sulfuric acid aerosol in mixed events (such as was present during the Raikoke time period) is not possible. To some extent we agree with this statement. For example, if an aerosol layer contains a mixture of smoke and sulfuric acid aerosol then we agree that SAGE is not able to determine the relative fraction of each component. Indeed, the proposed classification scheme would classify this layer as either sulfuric acid or smoke. Such an "either or" classification is not without its limitations, though it still provides scientific value. While the original manuscript provided a brief discussion on this topic, the revised manuscript has added sections to discuss misclassifications as well as to aid the reader in understanding the limitations of this approach.

Section 4. In this section K21 present a theoretical approach to understanding Vis-NIR extinction spectra for absorbing and scattering media. Smoke and sulfate are distinguished solely on the basis of the spectral variation of refractive index. This is interesting and informative. However, K21 state explicitly that their theoretical construct is "...in no way intended to be representative of actual conditions." This can be seen as a major weakness in that Figure 3b shows that an effective radius disparity between brown carbon and sulfate can be as small as  $\approx 10$  nm for an equal slope of - 1.5,  $\approx 25$  nm for -1.0, and 60 nm for -0.5. One can infer from the work of T21 (their Fig. 8) that the range of effective radius for a variety of sulfate plumes is quite large, perhaps exceeding 60 nm. Given the paucity of information on stratospheric smoke-plume and sulfateplume mode radius, and the reasonable expectation of systematic differences, it seems absorption systematics might be potentially inconsequential in the case of certain "actual conditions." In short, the theoretical experiment offers only one real-life particle-population systematic when others (to wit, particle size) will likely muddle plume distinction. If this paper is to rely strongly on theoretical underpinnings, the simulations must be multi-faceted.

We disagree that this is a major weakness of the methodology because this section presents the continuity of thought we applied in this analysis (from question to hypothesis to results) and Fig. 3 could be removed entirely from the paper without changing the analysis, results, methodology, or interpretation. Indeed, even without this figure the results from the 4 case study events would stand on their own. However, we believe that despite this model being quite simple, this figure is instructive in understanding the generalized pattern of what we observe. This figure has been updated to include more realistic BrC/BC mixtures and we have updated the discussion surrounding this figure. The 2 main points of the added information is: 1. BrC has a highly variable refractive index that may be similar to sulfuric acid or it may be almost as high as black carbon; 2. when a small amount of BC (10%) is added to the BrC simulation the spectral slope changed substantially. The BrC curve in the original manuscript was intended to act as a lowerlimit for smoke. Indeed, if a smoke layer is composed solely of BrC that has the refractive indices specified in Table 1 would be difficult to distinguish from sulfuric acid aerosol. However, if the smoke particles have a small contribution from BC and they have radii at the lower-end of the expected values for smoke, they are easily distinguishable from background sulfuric acid aerosol. This discussion was added to the manuscript.

Section 7.1. K21 state "...there were two pyroCb events in the northern hemisphere during the summer of 2019..." and cite Kloss et al. (2021), Vaughan et al. (2021) and Bachmeier (2019). Neither Kloss et al. nor Vaughan et al. provide concrete details of any pyroCb event; neither goes much farther than to claim that pyroCbs occurred in Canada and Russia. No information is provided on the massiveness of these injections, a crucial element. The Bachmeier citation refers to a blog post about a pyroCb in eastern Siberia on 30 April 2019. It is known, and can be gleaned from the Bachmeier post, that that pyroCb is highly unlikely to have been a major contributor to the stratospheric aerosol burden. Moreover, it occurred seven weeks prior to the Raikoke eruption. If this was considered a candidate for all the smoke K21 detected in 2019, it is incumbent on them to investigate that event much more deeply and quantitatively. The same goes for the Canada pyroCb event. K21 rightfully acknowledge that the pyroCb action in 2019 did not match noteworthy

pyroCb events such as the ones in British Columbia, 2017, and Australia in 2019/20. These, and a few others, were quantitatively massive, long lasting, and involved stratospheric plume lofting to the altitudes of the Raikoke plume in 2019. Presumably, any pyroCb that would have made a suitable contribution to stratospheric smoke in 2019 would be easy to identify. Relying exclusively on the vague information and citations provided herein is insufficient to buttress the extraordinary claims made by K21. In truth, there were in excess of 30 boreal pyroCbs in 2019. Some occurred prior to the Raikoke eruption, and several occurred thereafter. At least three were demonstrably large enough to create traceable intercontinentally transported plumes. Evidence of one of these plumes was inadvertently demonstrated by K21 in Figure 1. Hence, as in the Pinatubo summer of 1991, it is likely that stratospheric smoke was competing with Raikoke sulfates. But as argued above, this implies an obligation to apply much greater rigor in composition determination, necessarily involving several additional complementary data sets. SAGE data alone are insufficient. K21 are encouraged to either challenge that assessment or radically bolster their data analysis.

An example of the above suggestion was demonstrated by Cameron et al. (2021), who combined profile retrievals of SO2 and aerosol extinction in an examination of several stratospheric volcanic plumes, one of which was Raikoke. By exploiting coincident volcanic-gas and aerosol profiles, they presented a firstorder confirmation of sulfate particulate matter. In the case of Raikoke, Cameron et al. demonstrated close association of SO2 and aerosol enhancements in what K21 consider the "Raikoke Primary" and "Raikoke Secondary" plumes. This of course does not rule out some minor influence of biomass-burning-generated aerosol, but it clearly shows a picture of volcanic material over the altitude/zonal/temporal range examined by K21. Presumably, if K21's assertion of smoke-dominant presence is verifiable, complementary data embodying biomass burning signatures such as carbon monoxide would aid in identifying a sulfate/smoke blend. To make their case, K21 are encouraged to leverage data sets such as ACE-FTS and Aura Microwave Limb Sounder (e.g. https://acp.copernicus.org/articles/21/16645/2021/ in addition to CALIOP (exploiting its depolarization ratio data item).

We have reconsidered the interpretation of the Raikoke "secondary plume" that was observed over the tropics. We now demonstrate that smoke was detected betwen 25°N and 52°N up to  $\approx 20$ km as observed in the ACE-FTS data. We now discuss potential reasons for misclassifying the sulfuric acid particles in the secondary Raikoke plume as smoke (at higher altitudes) and present this as a limitation of this methodology and the importance of not limiting analyses to a single dataset.

However, we did demonstrate in the original manuscript that smoke was present in the higher latitudes within days of the Raikoke eruption. Further, Ansmann et al. 2021, Ohneiser et al. 2021, Osborne et al. 2021, and Johnson et al. 2021 reported observing smoke in the UTLS from  $50^{\circ}$ N to  $\approx 80^{\circ}$ N around the Raikoke time period. We see these studies as supportive of the claim that smoke was present in the northern latitude's UTLS around the time of Raikoke (up to 8-10 km). The manuscript was revised to include this discussion as well as a discussion on potential misclassifications.

Finally, we believe there is possible a misinterpretation of our results. When dealing with mixed events the classification scheme operate in an "either or" manner, which is a false dichotomy. In reality, these aerosol layers have a high probability of containing both aerosol types. Therefore, when a layer is classified as smoke this does not mean that that layer consists solely of smoke. It does indicate that this layer had enough smoke in it to push the classification into the smoke regime. This was discussed above, and this discussion is included in the revised manuscript for

**clarity.**

Table 3. Canada. The breakdown between sulfate and smoke shows that sulfates represent more than 12% of stratospheric aerosol enhancements at all altitudes, and dominate at 23-25 km. Whereas boreal pyroCbs occur every year, extratropical volcanic events of VEI=4 do not. To my knowledge, there was no 2017 boreal volcanic eruption. K21 do not cite any evidence of such an eruption. Thus, what is the rationale for assigning such an overwhelming number of SAGE observations to sulfate, especially at altitudes >22 km? Unless there is to be a claim of an unpublished, suspected volcanic eruption that year, it does not seem logical to categorize any SAGE measurements as sulfate. More logically, these are an indication of smoke that overlaps into sulfate extinction-spectra space, or simply uncertain. Please justify classifying any of the 2017 aerosol enhancements as sulfate.

We thank Dr. Fromm for pointing this out as this is a limitation of this methodology. A brief explanation of this phenomenon was in the original manuscript, but we provide a brief summary here. First, this is a simple classification scheme that depicts general patterns between smoke/volcanic events. Because the stratospheric background is primarily sulfate, this scheme is potentially biased toward sulfate identifications (i.e., it requires a enough smoke raise the extinction coefficient above the imposed statistical threshold for background conditions AND push the spectral slope over the imposed statistical threshold for background conditions). Therefore, we believe this is a conservative estimate of elevated smoke events. Second, where SAGE samples is important. It is possible that we sample through the edge of a plume wherein we observe enhanced extinction, but not enough smoke to push the slope over the statistical threshold. This would result in a false negative for identification of smoke in this example. While this was in the original manuscript, we have updated this discussion in the revised version.

Section 7.1, Lines 367-369. Indeed, it is true that there were no 2019 pyroCbs in the class of the 2017 Canada and 2019/20 Australian events. Then how to explain smoke rising to 25 km and lasting 7 months? Except for citing Kloss, Vaughan, and Bachmeier, no attempt is made to assess the magnitude of the 2019 pyroCb plumes to determine if they had the ingredients to generate such a lasting and self-lofting plume. It seems as if the authors are satisfied that the information presented in the cited works makes it self-evident. Either the 2019 pyroCbs had sufficient heft to exert the extraordinary impact reported by k21 or they didn't, in which case a novel interplay between sulfates and smoke occurred. K21 should undertake a more rigorous 2019 pyroCb survey and, depending on their finding, offer a cogent explanation for the processes that support their high-altitude, persistent smoke+sulfate anomaly.

We support the claim of identifying smoke in the mid-latitudes by including data from ACE-FTS. The objective of this manuscript is not to link these observations with specific biomass burning events, rather it is to evaluate a classification scheme for identifying smoke or sulfuric acid particles. That said, we agree that, above  $\approx 20$  km we see no support from ACE for the smoke classification. This is likely a misclassification due to large sulfuric acid particles that were present after Raikoke. We also note that we saw no support for the presence of ash within this secondary plume.

Line 436, 437. "Unfortunately comparison with CALIOP was not possible for the secondary

plume." K21 have missed a strategic opportunity to fully exploit CALIOP to help inform the sparser SAGE data at low latitudes. SAGE coincidences are not necessary determine the likelihood of smoke by way of CALIOP data. CALIOP depolarization ratio data are abundant from low to high latitude and throughout the life of the Secondary plume. It would be straightforward and advisable to analyze depolarization ratio for low- and high-latitude stratospheric layers to see if there is support for smoke.

It is true that a proximate match to a SAGE profile is required to make use of CALIOP data. However, this is not just a coincidence issue. The CALIOP signal at the corresponding altitudes of the secondary plume is quite low and making a reliable depolarization measurement at these altitudes is challenging. However, as noted above, we have updated our text to both support the claim for stratospheric smoke in this time period and to indicate that we see no support for the presence of smoke, at these latitudes, above  $\approx 20$  km.

Introduction, starting at Line 46. The next two paragraphs are interesting and well composed. But how relevant is this background to the issue at hand? I encourage k21 to prune the material here to improve the focus on the volcanoes in the satellite era.

Introduction has been updated.

Line124-126. What is the benefit of interpolating to 520 nm versus just adopting either 455 or 755 nm channels for analysis? I.e. what is special about 520 nm?

There is nothing is special about the 520 channel. There is a known issue in the 520, 601, and 676 nm channels as mentioned in the manuscript. While it is reasonable to expect that this channel, in the middle of the extinction spectrum, would play a relatively minor role in the slope calculation we did not operate under that assumption. Therefore, while building the analysis code we retained the 520 nm channel and explained our method of mitigating the aforementioned problem for this channel. However, we verified that excluding this channel does not significantly change the slope or subsequent analysis. The average of the absolute values of the percent difference (with\_520 - without\_520 / with\_520) was 0.03%, the median of the absolute values was 0.27%, and the 90th and 99th percentile of the absolute percent difference were 0.8% and 1.5%. The 520 nm channel could be dropped from the analysis without effect. This is now addressed in the manuscript.

Line 137-138. A citation is needed for the range of injection altitudes, especially "19 km." Citation added.

Line 138; callout of Fig. 1. Please see my previous comment about Fig. 1 and modify this sentence accordingly.

Figure 1 was updated and this sentence was updated to indicate the Raikoke plume was detected within 10 days as opposed to 1 week of the eruption.

Line 288-289. "Both depolarization ratio and the VFM were used herein to corroborate the identification of sulfuric acid aerosol and smoke within the SAGE data." That was apparently not done for Fig. 1. CALIOP near coincidences are available and show the requisite depol. ratio for

smoke. When reworking Fig. 1 and the attendant discussion, please include CALIOP coincidences.

Figure 1 now has a CALIOP backscatter profile and the granule is referenced. The TropOMI SO2 product is also shown in Figure 1.

Section 4. This is where K21 introduce the idea of a calculated slope as an alternative to extinction ratio. But I could not find where they precisely defined how slope was calculated. This would be essential for readers who would like to replicate K21's method and results. Please elaborate on the slope calculation.

Section 4 is where we introduce the hypothesis that slope can be used. The first paragraph in section 5 (Detection and classification method) describes how this is calculated.

Line 434, 435. The physical process described here is confusing and unclear. Smoke was shed from what? What is the cited precedent for smoke acquiring sulfate?

This section was updated in the revised manuscript.

**Response to comments within the manuscript. All line and figure references correspond to where Dr. Fromm's comment appeared in the original manuscript.**

Citation added to line 140.

Line 141: no, the colon (":") is the correct mathematical symbol for a ratio

Fig. 1: The tropopause was below 10 km for these figures (now indicated in the caption) and the line breaks are where the SAGE algorithm failed to produce valid data (now specified in the text).

Line 169: updated to specify wildfires were coincident in time.

Line 201: Changed "will have" to "may have"

Table 1: You are correct. The table has been updated.

Line232: corrected to read "three"

Line 235: The bias is mitigated using the power law. We have yet to implement a permanent correction for the 520 channel within the algorithm so we explain here that the impact of any unknown residual bias in the 520 channel should be mitigated.

Table 2: Altitude column was removed and the altitude range is now specified in the captions for Table 1 and Figure 4.

Figure 4: Yes, only data collected between 14-25 km were used (now specified in the caption).

Line 253: Updated to read  $2\sigma$

Line 313: This is a common Latin term for "see below". Common Latin phrases are allow by Copernicus (for example: e.g., i.e., et al. ad hoc, post hoc, vide infra, vide supra, in situ, etc.)

Lines 376–379 and Figure 14: You are correct that SAGE extinction showed values comparable to background. However, the slope profile indicated the presence of smoke, which was corroborated within the CALIOP depolarization ratio and VFM curtain plots. Regardless of the interpretation of the SAGE extinction data, the CALIOP data are unambiguous in their identification of smoke. The text has been updated to emphasize this.

Line 392: Comment already addressed in 2 location above.

Line 404: Text now updated to exclude comment about lifetimes.

Line 408: We stated that that hypothesis earlier in the paper. The manuscript was updated to indicate this.

Line 423: "near constant" is something that is not changing (i.e., second derivative is near 0) whereas "flattening" is a slope that is closer to 0 (i.e., first derivative is near 0).

We apologize, but we do not understand the issue with this part of the manuscript. Above  $\approx 1E-3$  (19 km) and  $\approx 7E-4$  (at 20 km) there is a clear divergence between the smoke and sulfuric acid designations. This is the "bifurcation" we alluded to in the manuscript; this is just a statement of an observation.

---

## Author Comment (AC3)

We would like to thank the third anonymous reviewer (AR3) for providing feedback on this manuscript. AR3 provided little in terms of suggestions or criticisms, rather the reviewer suggested a general agreement with the comments posted by Dr. Mike Fromm (AR2). We kindly ask the reviewer to see our response to Dr. Fromm as a general response to AR3's general comments. However, we address 2 issues raised by AR3 below. Our responses are provided below (red) to AR3's comments (black).

A shift of only about 50 nm in mode radius brings the two curves on top of each other. (Not even accounting for changes is distribution width or multi-modal, gamma, etc., shaped distributions) This means that BrC is essentially indistinguishable from slightly larger sulfuric acid droplets in terms of spectral slope when allowing for uncertainly in particle size.

There has been much confusion over the interpretation and utility of Figure 3. As stated in the original manuscript, there are many assumptions that go into creating this figure; therefore, this figure should not be considered representative of actual atmospheric conditions during any of the events presented herein and is not presented as a predictive model. Rather, this figure presents a *very generalized guide* for how particles of differing composition may change our measured extinction spectrum. Using this figure we developed the hypothesis that we *might* be able to distinguish between smoke/sulfate using the slope method. This figure does not *prove* that this is possible; rather, the case study events speak to this. It is true no information comes out of this figure for use in subsequent analysis; it is just a stepping stone in our original thought process and is presented to help the reader understand the general behavior of smoke and sulfuric acid particles. Indeed, this figure could be removed from the manuscript and the four case-study events would provide ample support on their own.

The 2 smoke curves show a range of potential values that are dependent on the composition (or degree of "complete" combustion) of the smoke. The actual refractive index for smoke is highly variable as shown be Liu et al. 2015 (now reference in the revised manuscript), and the refractive indices we chose provide a reasonable representation of the BrC RI *lower boundaries* in Liu et al. 2015's Fig. 4. As stated above, wildfire burns result in a mixture of BrC and BC being released into the atmosphere and the BC/BrC ratio will be highly variable depending on burn conditions. Further, the composition of BrC determines its spectral properties (i.e., refractive index), which results in a wide range of possible refractive index values (as now shown in the revised manuscript). Of course, this is all complicated by the lack of in situ measurements of stratospheric smoke. Indeed, it would seem that there is a great measurement and modeling opportunity here that should be seized, but is outside the scope of this manuscript. Regardless, what the revised Fig. 3 now clearly demonstrates is that a smoke particle that contains just 10% BC and has a nominal radius of 120 nm is easily distinguished from background sulfuric acid aerosol.

We updated this section to make these points clear to the reader and we than AR3 for pointing out this ambiguity.

However, figures 15 and 16 do not clearly show a sulfuric acid main (lower) peak and a smoke dominated secondary peak (unless I am missing something).

The reviewer is correct that there is not a clear partitioning of sulfuric acid and smoke as a function of latitude. We have updated our discussion of potential misclassifications within the

Raikoke data sets as well as added support for the identification of smoke from 25°N to 52°N, up to 20 km.

---

## Author Comment (AC4)

We would like to thank the fourth anonymous reviewer (AR4) for providing feedback on this manuscript. Our responses are provided below (red) to AR4's comments (black).

The authors did not discuss what the results of the classification method will, or could be, used for. Large wildfire smoke events may be possible to identify in periods of moderately volcanically elevated sAODs, but OMPS (UVAI, ext coeff) and Calipso (dep ratios, col ratios) already does this. Are there any advantage of using SAGE rather than other platforms?

While other instruments are routinely used to observe wildfire and volcanic activity, it is important to understand the applicability of SAGE data to these identifications. While coincident observations (in both time and space) strengthen the interpretation of each instrument's data, it is not always possible to have multiple instruments observe the same location within a reasonable time frame. Therefore, it is necessary to be able to understand the strengths and limitations of each individual instrument. To this end, we presented and evaluated a method of distinguishing between sulfuric acid aerosol and smoke in the stratosphere that uses the SAGE extinction spectra. The introduction was updated to communicate this.

Can you be sure that it is only wildfire smoke and not something else? Carbonaceous components have been found in volcanic aerosol. Could it not be that one of the eruptions of Raikoke contained soot and organics, or that smoke in the area of Raikoke was entrained in the volcanic cloud?

We apologize, but we do not understand what part of the manuscript this comment refers too. If this is just a general comment, then our response is: We discuss in the paper why some misclassifications may take place, though we see no evidence for soot and/or organics being a significant component of the Raikoke ejecta. The misclassification discussion has been updated in the current manuscript.

Further, I found no difference in depolarization ratio for the assumed wildfire smoke and volcanic aerosol (after Raikoke). Why does it not show up as wildfire smoke in calipso's depolarization ratio? You show information on particle sizes, but not on other particle properties. I think that the depolarization ratio should be shown here since it is a very strong indicator of smoke.

We apologize but we do not understand what the reviewer is referring to. If this is in reference to Figure 14, this figure demonstrates that smoke was in the stratosphere prior to the Raikoke eruption.

Section 5: In the beginning of this section, it reads that the slope was computed via linear regression. How did these regressions look and how well did they fit to the data? From the figures, e.g. Fig4, it looks like there is a large variance in the data. It is difficult for the reader to grasp this without some type of illustration of these regressions. Aren't the widths of these distributions rather important for your classification? The standard deviations of these linear regression models could be used to distinguish between cases where the identification is more or less 100% indicative of one class, and cases where the data points are mixtures of smoke and sulfate. I think that this could be a means of telling whether the rising stratospheric aerosol after Raikoke/Ulawun/Fires are a mixture of smoke and sulfate or only smoke.

It is correct that slope was computed via linear regression, though the residuals of this fit were not part of the calculation and subsequent analysis. There may be some confusion here. Figure 4 used extinction ratios and not slopes. This figure is not part of the analysis, but was a preliminary step we used to observe the general behavior of the SAGE data to see, to a first approximation, whether the data behaved as expected from theory (i.e., based on Fig. 3).

Regarding the "standard deviations of these linear regression models", again, we apologize but we do not see how standard deviation of a linear regression plays a role here.

I think that the figures showing the altitude dependent slopes are really good illustrations to highlight where the different aerosol layers are located. Does this work well when separating background aerosol from low volcanic impact?

As demonstrated by the case-study events this works well when the criteria on lines 266-272 are met, as described in the original manuscript. Ultimately this depends on how much the "low volcanic impact" events change the extinction coefficient.

In the analysis you had to divide data into altitude segments (since extinction coefficients in rising or descending air masses becomes pressure dependent). Would it be possible to normalize the data with pressure to get an altitude independent slope for each class (backgr, volc, smoke)?

Data were broken into altitude ranges because the background aerosol load changes as a function of altitude. Therefore, we had to develop an altitude-based statistical set for each location to determine whether or not a plume was enhanced). This statistical set may be used at different locations within the same latitude band, but the altitude grid must be retained. Broad application (in both altitude and latitude) of a statistical data set is ill advised in this scenario.

Table 3: It looks to me that there are quite some misclassifications even at times and altitudes with large sample sizes. Starting with the Canadian fires 2017, 62% of the data at 14 km altitude were classified as sulfuric acid, and at the highest altitudes (23-25 km) 57-99% are misclassified as sulfuric acid. What would be the source of this sulfuric acid? I don't know of any potential eruptions occurring in the first half of 2017. To me this indicates big issues with the assumptions used for the classification algorithm. The same issue occurs after the Australian fires 2019/2020, but only at the highest altitude shown (25 km). I would like to see how well the algorithm does above 25 km. It is evident in the SAGE 3 iss data that the smoke rose to >30 km, and some dense smoke layers in the v5.10 data lacked data below 27 km indicating too high optical depth in the line of sight to quantify the extinction. So these are not faint layers. It is difficult to interpret the classification results after Raikoke if these issues occur in the periods of known sources.

We do not propose a sulfuric acid source for these time periods. Rather, we discussed potential reasons for misclassification of these very events from lines 334-354 of the original manuscript. This discussion has been expanded in the revised version.

It is true that smoke was detected to high altitudes after the Australian fire. However, we limited the scope of our analysis, as presented here, to 14-25 km for 2 main reasons. 1. we used extinction coefficients from 450-1550 nm in this analysis to calculate spectral slope. The sensitivity of longer-wavelength channels decreases rapidly above 25 km. 2. Considering the case study events as a group, there were fewer interesting enhanced layers in this altitude regime.

After Raikoke all the highest extinction coefficients (Fig 15) were classified as smoke. I find this surprising. Does this mean that a large fraction of the AOD elevation after Raikoke was actually caused by fires?

This is an excellent point and we thank the reviewer for raising this issue. No. Just because a layer is identified as "smoke" does not mean it is composed 100% of smoke, and it does not mean that that layer is majority smoke. This is important and we thank the reviewer for raising this question. We have added additional comments regarding this interpretation and the potential for misclassification to the text.

I wonder to what degree the small difference in refractive index affects the classification. The refractive index for black carbon differs quite from that of H2SO4. However, brown carbon and sulfuric acid has rather similar values in refractive index, indication that it is difficult to separate between the two.

The 2 smoke curves show a range of potential values that are dependent on the composition (or degree of "complete" combustion) of smoke. The actual refractive index for smoke is highly variable as shown be Liu et al. 2015 (now reference in the revised manuscript), and the refractive indices we chose provide a reasonable representation of the BrC RI boundaries in Liu et al. 2015's Fig. 4. As stated above, wildfire burns result in a mixture of BrC and BC being released into the atmosphere and the BC/BrC ratio will be highly variable depending on burn conditions. Further, the composition of BrC determines its spectral properties (i.e., refractive index), which results in a wide range of possible refractive index values (as now discussed in the revised manuscript). Of course, this is all complicated by the lack of in situ measurements of stratospheric smoke. Indeed, it would seem that there is a great measurement and modeling opportunity here that should be seized, but is outside the scope of this manuscript. This figure was updated to show how addition of a small amount (10%) BC significantly changes the overall slope as compared to the pure BrC curve. The consequence of this is that if a smoke plume is composed of 90% BrC and 10% BC and has a nominal mode radius of 125 nm (on the lower end of what is expected for smoke particles) then the resultant slope is easily distinguished from the slope yielded by background sulfuric acid aerosol. This explanation has been added to the manuscript for clarity as has additional language on the use of this figure as a predictive model. This figure presents a general relationship and while we expect this general relationship to be valid, we explicitly state that stratospheric smoke is more complicated than this simple model and that the case study events stand on their own, independent of this figure. Further, the manuscript now contains expanded discussion on the possibility of misclassifications.

Why did you not include a spectral slope for ash (Fig. 3), and in what way may this impact your classification?

Much like smoke, ash has a highly variable composition that results in variable refractive indices. However, we did discuss in the original submission that ash should behave like smoke due, primarily, to it's larger size. As shown in Fig. 3 of the original manuscript, as particle size increases the spectral slope tends to flatten. Therefore, we expect it to have a flatter slope than sulfuric acid aerosol, and will therefore be misclassified as smoke in the current method.

The numbers in Table 3 don't add up. I did not check them all but noticed the issue at

Australia @25 km altitude (0.30 + 0.60)

We are appreciative of the reviewer's keen eye and pointing this out. This was corrected.

L325-328, regarding Fig 7&8: You claim a rapid decrease in the slopes. I see a slope changing value over several kilometers. Isn't this an indication of mixed sources?

The slopes for the plumes indicated in the text changed rapidly over an altitude range of 2.5 km (Fig. 7) to 1 (Fig. 8) km. Similar enhancement in extinction coefficient is observed in panel (c). While the width of the plume is ≈5 km thick, the rate of change from background conditions was high. To the reviewer's point: we see no indication of a mixed source in the Ambae and Ulawun events.

Regarding the figures with calypso curtains, I suggest that you add curtains of the beta-532 signal as it is difficult to understand why there is a yellow feature in Fig 7b (same in Fig 8). The volcanic layers should be visible in beta-532.

We thank the reviewer for this suggestion. CALIOP backscatter products were added to these figures.

L358: You write about an aerosol layer at 19 km altitude. It is actually visible in calypso images, but it is classified/misclassified as clouds. No cirrus should be present so far ( 7-8 km) above the TP (even above the ExTL).

The reviewer is correct that a layer was identified in the CALIOP VFM ≈5 to the north and ≈7 to the east of the SAGE overpass location, but it was not visible where SAGE sampled the atmosphere.

Section 2.1: Why was the data limited to 2 km above the TP? Was it to minimize cloud interference? And why not 1 or 3 km? Are there any risk of cloud interference that may disturb the classification?

Correct, this limited the impact of cloud interference. While cloud interference is always a concern near the tropopause, we see no evidence of cloud interference in the data within the selected range.

Section 2.2: The lvl3 sAOD product has strong bias in the extratropics (Kar et al. 2019). Does this have any impact on the comparison with SAGE?

We apologize, but we do not understand the intent of this comment. The L3 CALIOP sAOD product was not used in this analysis.

Caption Figure 3: I think that you should add the word 'normalized' to the ylabel.

We appreciate the suggestion, but this information, in addition to an explanation of how normalization was done, is in the caption.

There is strong gradient in the slope in Fig. 12d at 10-11 km altitude. Could it be clouds interfering? Also, no TP was marked in Fig.7&8. Is the TP height lower than what's shown in the graphs?

No, this is well above the tropopause ( 8 km), which can be seen in panels (a) and (b) of these figures.

The particle size distribution evolves with time, especially in the first month or two after eruption (or smoke injection). This should lead to increased variance in your data.

We thank the reviewer for this comment, but we do not see anything actionable from this.

---

## Referee Report (RR1)

**Review of Knepp et al. (AMT-D, 2021), "Identification of Smoke and Sulfuric Acid Aerosol in SAGE III/ISS**
**Extinction Spectra Following the 2019 Raikoke Eruption"** *- Revised manuscript*

**Reviewer: Mike Fromm**

**K21 have made substantial and necessary changes to their manuscript. Even though they are necessary, they are not sufficient to alter my major concerns with the paper. The root of my concerns is that their method of inferring particle composition is based wholly on the spectral slope of aerosol extinction in the visible to near-IR and that they provided no independent observational proof that wavelength-dependent aerosol extinction in this range unambiguously carries particle-composition information. K21 appropriately present the two main elements that might distinguish smoke from sulfates in their imprint on vis-NIR extinction spectra, absorption and particle-size distribution (PSD). However, a priori establishment of these two qualities is not satisfactorily made. The role of absorption is presented in the theoretical realm, embodied in Figures 3 and 5. But the confounding effect of PSD is given inadequate attention. If there is to be a single, algorithmic relation between aerosol extinction and spectral slope, it must be established that volcanic sulfates and wildfire smoke have characteristic, systematic differences in PSD. If there is no such systematic difference between smoke and sulfate, then absorption vs. scattering will rule and there should then be an unambiguous systematic difference on the spectral slope between clearly homogeneous smoke and sulfate populations. The evidence provided in K21's tabular results shows that this is not the case. If in fact PSDs between smoke and sulfate are substantively different, then this factor should be presented as an empirical determinant on the spectral-slope construct (Figure 3). However, to my reading, K21 do not make that case.**

**A more specific major concern I expressed in the original review relates specifically to the PSD issue. I pointed out that Thomason et al. (2021) (T21) showed that even within the superset of presumed volcanic sulfates, vis-NIR extinction spectra vary widely among diverse volcanic events. Although K21, in their response to my review, rightly argue that T21 did not consider possible composition mixes, the large range of spectral extinction variation among plumes that are almost certainly exclusively volcanic (i.e. the plumes from tropical eruptions) speaks to a very wide constraint on pure-sulfate PSD. Raikoke falls within the range of these tropical sulfate plumes (T21's Figure 8, below), hence it is a considerable challenge to argue that there is a motive for questioning the Raikoke-year's composition mix considering T21.**

[Figure]

**Figure 8.** The "before" (left-hand point) to peak 1020 nm aerosol extinction coefficient (right-hand point) for the 10 eruptions considered in this study is shown in panel (a), and the differences between them (perturbations) are shown in panel (b).

K21 assert in their response to my review that T21's analysis has a different motive and construct as compared to K21, but extinction ratio and a spectral slope derived therefrom are inextricably linked to the same underlying determinant: small particles yield both a large extinction ratio and spectral slope.

It is not logically clear why K21 make an exception for Pinatubo, when inferred volcanic sulfate PSD covers a quasi-continuous range from small to large (T21). Indeed Raikoke is situated deeply into the large-particle side of the PSD spectrum according to T21. Moreover, Thomason (GRL, 1992) characterized Pinatubo aerosols as the norm for volcanic clouds such that the "new mode" found within SAGE II extinction spectra was labelled "new" and hypothesized to be a transitional mode in volcanic sulfate-particle growth. To the extent that pyroCb smoke explained the Thomason (1992) "new mode," its relevance to the current manuscript is central to my assessment that vis-NIR extinction spectra are inadequately constrained for particle-composition inference. To wit, had K21's spectral slope algorithm been applied to Pinatubo, it is likely that SAGE Pinatubo measurements would have been classified as smoke and pyroCb smoke measurements would have been closer to sulfates.

The point I made in my first review about T21 was simply that their illustration in Figure 8 was sufficient to caste large doubt on any attempt to derive a binary choice between stratospheric smoke and sulfates. T21's "perturbation" 520/1020 nm extinction ratios are all over the place, from less than background to the upper limit: unity. Roughly half the plumes have increased extinction ratios (more negative slopes, in K21 parlance) w.r.t. background, the remainder have smaller extinction ratios (less negative slope). Raikoke is in a class with three other plumes with decreased extinction ratio, each created by a tropical volcano and no question as to contributory smoke. The different motives and methods of T21 vis a vis K21 are not germane. The patterns within T21 Figure 8 map directly to K21. The fact that T21 only consider the background and peak perturbation in Figure 8 serves to clarify my point, not

diminish it. The T21 perturbation state represents extinction signals at absolute and relative maximum, a state of minimum uncertainty in the signal. Hence it is a demonstration of the optimal circumstances for assessing both microphysical evolution and composition. Hence, T21 Figure 8 shows that there is no firmer footing for questioning Raikoke's composition mix than for questioning that of any of the other three plumes in Raikoke's class.

Still on the topic of T21, K21 state in the manuscript and their referee response that Pinatubo cannot be addressed with their approach. Their argument is understandable, but not compelling. Of course, particle populations of a size class that would render SAGE color ratios (spectral slopes) near unity (near zero) would inhibit interpretation. This situation is illustrative of a realm where the most appropriate finding is "uncertain." Because population-scale particle size, type, and blend fall naturally on continuous scales, so too does their manifestation in the vis-NIR extinction realm. Just as there is relatively large uncertainty in composition at the near-background level of extinction, there is also gradually increasing uncertainty as extinction ratio approaches unity. In accordance with this perspective, the natural suggestion is to define an uncertainty category in the composition algorithm rather than descoping particular volcanic plumes. The case of Pinatubo is illustrative. One can see from T21's Figure 7 that there is a point in the SAGE sampling of the young plume during which color ratios are not pegged at 1. If one considers that point in the plume's evolution, the extinction/extinction-ratio pattern closely resembles that of Kelut in T21 Figure 7 (accounting for the different ordinate scales in panel 7a and 7c) and 8. In summary, the construct presented in T21 bears two lessons for K21. One is that the Raikoke-era plume is indistinguishable from several presumably sulfate-only plumes. The other is that it is necessary for an algorithm aimed at discerning composition (or blends) needs to go beyond a smoke/sulfate binary outcome.

At the risk of "beating a dead horse," I offer one other argument related to T21. I took the example of Kelut (February, 1990) to illustrate how this volcanic cloud could suggest a smoke blend, according to K21's construct. A basic rendering of 1-micron extinction and 525/1020 extinction-ratio as a function of time are shown below. The data are version 7 SAGE II. One representative retrieval altitude (19 km) makes up this example. The extinction data are limited to profiles between 0-20°S latitude.

[Figure]

Next is an analysis akin to K21's Figure 4: color ratio as a function of 1020 nm extinction. It is clear that this distribution of background color ratio to perturbation color ratio is more like the two smoke examples in K21 Figure 4 and the two sulfate examples. Undoubtedly, Kelut's inarguably sulfate plume at 19 km would have been in the smoke category within the K21 construct (modified as it would be for SAGE II wavelengths). Not shown, but evident at other stratospheric altitudes up to 21 km, is this same pattern. At higher altitudes it morphs to a more K21 sulfate pattern. This is reminiscent of the altitude variation of smoke/sulfate proportion systematics presented in K21. To me this is further evidence of the weakness of the SAGE data and the K21 construct for composition inference.

[Figure]

In response to referee suggestions, K21 invoked additional satellite data to qualify their findings. For instance, they showed TropOMI SO2 in concert with a SAGE profile (Figure 1) deemed to sample sulfates. This was very helpful. However, it would help the reader if K21 would refer to Figure 1 and report on the algorithmic result for this profile (as they did of another example shown in Figure 19). At a glance, the 520/1550 extinction ratio at peak extinction is quite small (~6.7) in comparison to the two sulfate distributions in K21 Figure 4 (~10 being the minimum) and much closer to the two smoke examples.

They also introduced ACE-FTS gas and aerosol information. In this endeavor, they cited Chris Boone of the ACE-FTS team for a still novel aerosol-infrared spectrum construct for aerosol-type determination. While this led to improvements in the manuscript, in my opinion K21 drew a conclusion regarding the secondary and primary Raikoke plumes that fall short of convincing. For instance, K21 state that they find no ACE evidence of smoke above 20 km, using Figure 18 as an illustration. Absent though is any discussion of whether this conclusion is also informed by other ACE profiles not shown. Did K21 find, for instance, no CO or HCN enhancements above 20 km? Did they find a substantial number of ACE profiles with such enhancements up to 20 km? If so, that would bolster their conclusion. Specific to Figure 18, there is a concern with respect to the profile in which the ACE aerosol-infrared spectrum approach showed a smoke signal at 20 km (Figure 18 j,k,l). The smoke marker is at 20 km, above any CO or HCN enhancement, but centered on a huge SO2 enhancement. At a glance this appears to be confusing at best, erroneous at worst. Given that this profile is at 24N, 34E (situated on the western side of the Asian Summer Monsoon anticyclone, it may be a case of upper tropospheric biomass burning signal below the SO2 peak, and a volcanic plume above.

It should be examined more closely before a final determination, but as it stands, it is not a compelling foundation for K21's conclusion that smoke was observed up to 20 km.

Regarding the stratospheric Raikoke-season plume that ascended to 22+ km at low latitudes, what K21 refer to as the "secondary" plume, it is evident that the smoke angle and this anomalous plume height are a prime motivation for this paper; it is highlighted in the abstract. The original and revised K21 manuscript provide an unclear motivation to pursue this as smoke assisted. In the Introduction, K21 state: "The working hypothesis was that this secondary plume consisted, at least in part, of wildfire smoke." The "was hypothesized" part implies that the idea preceded K21 yet no citations are given. Also in the Introduction, they cite 3 papers on the subject of diabatic smoke-plume lofting, but none of these involve the Raikoke-season plume. I did not find any attribution of this hypothesis to the Raikoke secondary plume" cited papers. If I missed it, the authors are asked to provide the earlier sources. Otherwise, the "working hypothesis" should be claimed as their own.

Related to the above point, the original manuscript was criticized for implicating "2 major wildfires" (Abstract) as contributors to stratospheric aerosols in 2019. The criticism was that the manuscript relied on citations that provided only vague and inconsequential support for the wildfire/pyroconvection angle. I argued that the fires and pyroconvection in 2019 were not "major" in comparison to a normal pyroCb season (while acknowledging that stratospheric smoke injections were an annual occurrence). Yet the current manuscript retains these arguments. My challenge to K21, to establish a strong argument for wildfire/pyroconvection influence, went largely unmet. That challenge remains and is an essential component for convincing the reader that fires such as those in 2019 could independently bolster the argument for a wholesale pollution of the stratosphere in the Raikoke season. I have a suggestion, to look at a pyroCb season such as 2018 with SAGE data. In that year an equal number of pyroCbs occurred in the northern hemisphere as 2019, but in the absence of a volcanic cloud and the demonstrably huge smoke plumes in the NH in 2017 or the Australian plume in 2019/20. If a sizable smoke presence were to be found, it would bolster their argument that a normal pyroCb season (such as 2019's) could present a blend worth considering in the Raikoke year.

---

## Referee Report (RR2)

Review of Knepp et al., 2[nd] Revision)

Reviewer: Mike Fromm

The authors have once again made enormous changes and improvements to their manuscript. These are greatly appreciated, and allow me to acknowledge acceptance for publication after consideration of a few very minor suggestions.

Suggestion: reach out to Chris Boone regarding the ACE-FTS occultations presented herein, especially the two in July. Dr. Boone has dropped those from his current manuscript, under review, on Raikoke aerosols. This is largely because of uncertainties he had with the smoke spectra.

Section 6.4.4. The author's conundrum regarding Figure 20 and the generally assumed SO2-sulfate conversion timeframe of ~30 days has some particularly relevant reinterpretation regarding sulfate abundance much sooner post eruption. Guo et al. (2004; doi:10.1029/2003GC000655) discussed observations of very young sulfates in the Pinatubo cloud. De Vries et al.(2014; doi:10.5194/acp-14-8149-2014)  documented

stratospheric sulfates in the active Nabro eruption umbrella cloud and in the days immediately after the 13 June 2011 eruption. In addition, two works have shown substantial Raikoke stratospheric aerosol optical depth well within the first month after eruption (Kloss et al. 2021; Gorkavyi et al. https://doi.org/10.5194/amt-14-7545-2021). And since CALIOP is invoked in Knepp et al., it might be worth reviewing these data to show that the Raikoke SO2 cloud was imbedded with aerosols each day after the eruption. While some of these observations might be interpreted as ash, the important take-away is that there were indeed Raikoke particles in the stratosphere from the get go. Knepp et al. give a very nice detailed example of Raikoke sulfates on 2 July, which were imbedded in a synoptic-scale SO2 plume. Here, CALIOP provides a larger context for the SAGE layer, with signals interpreted as sulfate. https://www-calipso.larc.nasa.gov/products/lidar/browse_images/show_v4_detail.php?s=production&v=V4-10&browse_date=2019-07-02&orbit_time=09-10-53&page=1&granule_name=CAL_LID_L1-Standard-V4-10.2019-07-02T09-10-53ZN.hdf

---

## Author Response (AR3)

**1 Response to Reviewer #1**

We would like to thank Mike Fromm once again for providing feedback on this revised manuscript. We also appreciate Dr. Fromm's willingness to engage in fruitful conversations about this manuscript via e-mail and telephone. Our responses are provided below (red) to Dr. Fromm's comments (black).

The authors have once again made enormous changes and improvements to their manuscript. These are greatly appreciated, and allow me to acknowledge acceptance for publication after consideration of a few very minor suggestions.

We heartily appreciate Dr. Fromm's acknowledgement of our efforts to improve this manuscript.

Suggestion: reach out to Chris Boone regarding the ACE- FTS occultations presented herein, especially the two in July. Dr. Boone has dropped those from his current manuscript, under review, on Raikoke aerosols. This is largely because of uncertainties he had with the smoke spectra.

We have confirmed with Chris Boone that the data presented in these figures is accurate and the interpretation, as presented, is correct.

Section 6.4.4. The author's conundrum regarding Figure 20 and the generally assumed SO2-sulfate conversion timeframe of ≈30 days has some particularly relevant reinterpretation regarding sulfate abundance much sooner post eruption. Guo et al. (2004; doi:10.1029/2003GC000655) discussed observations of very young sulfates in the Pinatubo cloud. De Vries et al.(2014; doi:10.5194/acp-14-8149-2014) documented stratospheric sulfates in the active Nabro eruption umbrella cloud and in the days immediately after the 13 June 2011 eruption. In addition, two works have shown substantial Raikoke stratospheric aerosol optical depth well within the first month after eruption (Kloss et al. 2021; Gorkavyi et al. https://doi.org/10.5194/amt-14-7545-2021). And since CALIOP is invoked in Knepp et al., it might be worth reviewing these data to show that the Raikoke SO2 cloud was imbedded with aerosols each day after the eruption. While some of these observations might be interpreted as ash, the important take-away is that there were indeed Raikoke particles in the stratosphere from the get go. Knepp et al. give a very nice detailed example of Raikoke sulfates on 2 July, which were imbedded in a synoptic-scale SO2 plume. Here, CALIOP provides a larger context for the SAGE layer, with signals interpreted as sulfate.

There are multiple competing factors involved. One of which is a zonal coverage of large particles (>180 nm) within days of the eruption. This is not expected, so we are comfortable leaving the text as is. However, Dr. Fromm is correct that we must acknowledge the fact that sulfuric acid particles do form rapidly post-eruption as indicated by the provided references. Therefore, we updated this section (specifically point #1) to communicate this important aspect.

**2 Response to Reviewer #5**

We would like to thank this anonymous reviewer for providing feedback on this revised manuscript. Our responses are provided below (red) to the reviewer's comments (black).

The authors now fairly openly and critically mention the possible issues with the applicability

of the method to distinguish smoke from sulfate particles and the manuscript has improved significantly in my opinion. I also want to mention that the manuscript is very well written and easy to follow.

We appreciate the reviewer's comments and the effort in reviewing this manuscript.

There is one general aspect that should be discussed explicitly in more detail (parts of it are already discussed) in my opinion: your method will not work for volcanic eruptions that lead to larger sulfate particles (this is mentioned in the paper). Your approach to deal with this is to assume that weak eruptions (like Ambae and Ulawun) with relatively small sulfur amounts injected into the stratosphere will not lead to larger sulfate particles, but larger eruptions (like Raikoke or also Pinatubo) will. While this may well be possible, it is in my opinion not well established. The realization that volcanic eruptions will not lead to an increase in particle size (or will even lead to a decrease) is a very recent one and to my knowledge it is not yet fully understood what processes or parameters determine, whether a volcanic eruption will lead to smaller or larger sulfate particles. The SO2 amount may well play a role (it probably does), but perhaps also the injection altitude (and hence temperature of the ambient air). Certainly the relative roles of nucleation of new particles and condensation onto existing particles will be important. But I would not – considering the current level of understanding – exclude the possibility of a weak eruption that also leads to larger sulfate particles. The Kelut 1990 eruption may be such an eruption (as discussed in Thomason et al. 2021), although there may be issues with volcanic ash in the weeks immediately after the eruption.

To summarize: I think it would be appropriate to add another disclaimer stating that – while it appears plausible – the SO2 amount injected into the stratosphere may not be the only parameter determining whether the sulfate particles will become larger or not.

You are correct that injection mass alone is insufficient to predict particle size. However, if particles become smaller after an eruption then the proposed classification algorithm should correctly identify these particles as sulfuric acid (the corresponding slope will be more negative than background). Regarding the 1990 Kelut eruption, you are correct that the appearance of large particles was likely due to ash. However, as Thomason et al. 2021 demonstrated the background particle load can influence the growth rate of particles significantly. This is already discussed in the manuscript, so we see little to change. However, we now explicitly state that injected SO2 mass alone is not a good predictor of resultant particle size the "Application to the Raikoke Event" section.

I have two more general, but rather minor comments:

1. When you speak of particle radius you usually use the term "mode" radius and I'm wondering, whether this is the intended term? In the standard formulation of a log-normal distribution the variable $r_m$ or $r_0$ is the median radius, not the mode radius. This affects the text and also some of the Figures, e.g. Figures 1, 4, 5 and others.

We agree that this terminology can be confusing. Unfortunately the aerosol literature is highly inconsistent on naming conventions as well as assigning variable names. This is one such case. The use of "mode radius" is one such case. This term refers to the second mode of the distribution, therefore we use this terminology within this manuscript. However, we agree that this may cause unnecessary confusion and have updated the text to clarify this point.

2. The "§" sign is used frequently to refer to different sections and subsections. I think this is not standard Copernicus terminology and suggest using "section" etc. instead.

*We thank the reviewer for pointing this out and we have updated the text to remove the symbol and use "Sect.", per the Copernicus guidelines.*

Specific comments:
Line 369: "The reason for the presence of elevated sulfuric acid aerosol"
We don't know if they were present, right? Perhaps: "The reason for the potential presence .."

*Updated per the reviewer's recommendation.*

Line 50: Please provide references for the 30+ Tg of SO2 and 1 K global temperature perturbation. The latter value seems a bit large to me. Also, the injected SO2 mass is higher than most estimates I am aware of.

*I think you are correct that most estimates are around 20 Tg SO2 and 0.5–1 K. The original text represented upper limits (i.e., "upwards of"), but to remove ambiguity I changed the text to represent a 20 Tg SO2 injection and a temperature change between 0.5 and 1 K. I also added references.*

Line 162: "While the real component of the BrC refractive index is spectrally flat, it ranges from 1.3 to 1.9"
With "spectrally flat" you mean: no spectral dependence, right? The range from 1.3 to 1.9 does not refer to a spectral change, but to a dependence on composition? Perhaps this can be mentioned explicitly?

*That is correct, no spectral dependence. We agree this is confusing and have updated the text to: "While the real component of the BrC refractive index has no spectral dependence, previous studies reported refractive indices over a relatively broad range: between 1.3 to 1.9"*

Line 185: "this model is provides" -¿ "this model provides"
*Updated to the reviewer's recommendation.*

Figure 1: Definition of the slope. You carry out a linear regression with wavelength on the x-axis and log(k) on the y-axis, right? Log(k) is dimensionless and therefore the units of the slope should be: 1/nm, right?

*We believe you are correct. This has been updated throughout the paper.*

Line 258: Your criterion 2 will automatically exclude the possibility to have larger sulfate particles after a volcanic eruption. This is of course discussed in the paper, but in my opinion this still is a weakness of the approach, because research on the change in particle size after volcanic eruptions is still ongoing and the realization that sulfate particles may become smaller after eruptions is a very recent one.

*If particles become smaller then the slope will become more negative, which will force the*

classification to remain sulfuric acid. We interpret this comment as just a comment and see nothing to change here.

Line 282: "In theory, the proposed classification method is straight forward and is expected to be reliable for events of a single type (i.e., either volcano or wildfire, but not necessarily mixed events)."
Only if the sulfate particles do not become larger after a volcanic eruption, right? This should probably be stated here.
This is correct. The text was updated to reflect this.

Lines 332-335: The processes may be more complicated than suggested here. An important question is, whether nucleation (of new particles) of condensation (onto existing particles) is the main sink for gaseous sulfuric acid. I'm not sure there is a general answer and the answer may depend on injected SO2 mass, injection altitude (i.e. ambient temperature) etc.
We agree with the reviewer that this is poorly understood. We see no corrective action to be taken here.

Line 334: "its impact on the spectral slope was minimal due to the consistent composition and hence spectral properties"
This sentence implies that the spectral properties depend on the composition only. However, they also depend on the particle size.
This was updated to reflect the importance of the particle size distribution.

Line 341: "vide infra"
Perhaps in english? Not all readers will be familiar with this latin expression
Perhaps, but there is no harm in learning something new. We will abide by the guidance provided by Copernicus.

Line 345: "as was seen during Pinatubo." I don't fully understand the implication of this part of the sentence. Do you mean that the spectra were flatter after Pinatubo because of ash or because of large sulfate particles?
This is in reference to the sulfate particles. The Pinatubo reference here is superfluous so it was removed for the sake of clarity.

Table 5: This comparison of particle sizes is interesting, but should be complemented by a bit more information. It is not clear, whether this is an apples-to-apples comparison, because the assumed particle size distributions may be different. You assumed a mono-modal log-normal distribution (Fig. 1) with a width of 1.5. What was the width parameter for the Wrana retrieval? What size distribution was assumed for the ACE-FTS retrieval? Perhaps it makes sense to determine and present effective radii, too. This may allow for a better comparability of the values.
This is a fair point. Both the Wrana and ACE-FTS methods fit the SAGE and ACE-FTS data,

respectively, to the best matching particle size distribution. Therefore, there is no assumption of size distribution parameters (i.e., unlike us, they do not assume the width). You are correct that width can play a role in the extinction spectrum, though generally not substantial over small changes. We now include the sigma estimates in this table for clarity.

Are you comparing your "mode" radius to the "median" radius by Wrana? Or are you also using the "median" radius?

As stated above there is variability in how different people communicate the mode radius/median radius. It's the same thing, but can be confusing with the statistical mode. This has been clarified in the manuscript (see above). Felix and I both use the same value, though we call it by different names.

What's also not entirely clear: did Felix Wrana provide the retrievals or did you implement the Wrana method and carry out the retrieval yourself?

Felix Wrana provided these data. The text has been updated to make this clear.